# GFlowNets and variational inference

**Nikolay Malkin**∗**, Salem Lahlou**∗**, Tristan Deleu**∗
Mila, Université de Montréal

**Xu Ji, Edward Hu**
Mila, Université de Montréal

**Katie Everett**
Google Research

**Dinghuai Zhang**
Mila, Université de Montréal

**Yoshua Bengio**
Mila, Université de Montréal, CIFAR

## Abstract

This paper builds bridges between two families of probabilistic algorithms: (hierarchical) variational inference (VI), which is typically used to model distributions over continuous spaces, and generative flow networks (GFlowNets), which have been used for distributions over discrete structures such as graphs. We demonstrate that, in certain cases, VI algorithms are equivalent to special cases of GFlowNets in the sense of equality of expected gradients of their learning objectives. We then point out the differences between the two families and show how these differences emerge experimentally. Notably, GFlowNets, which borrow ideas from reinforcement learning, are more amenable than VI to off-policy training without the cost of high gradient variance induced by importance sampling. We argue that this property of GFlowNets can provide advantages for capturing diversity in multimodal target distributions.

Code: `https://github.com/GFNOrg/GFN_vs_HVI`.

## 1 Introduction

Many probabilistic generative models produce a sample through a sequence of stochastic choices. Non-neural latent variable models (e.g., Blei et al., 2003), autoregressive models, hierarchical variational autoencoders (Sønderby et al., 2016), and diffusion models (Ho et al., 2020) can be said to rely upon a shared principle: richer distributions can be modeled by chaining together a sequence of simple actions, whose conditional distributions are easy to describe, than by performing generation in a single sampling step. When many intermediate sampled variables could generate the same object, making exact likelihood computation intractable, hierarchical models are trained with variational objectives that involve the posterior over the sampling sequence (Ranganath et al., 2016b).

This work connects variational inference (VI) methods for hierarchical models (i.e., sampling through a sequence of choices conditioned on the previous ones) with the emerging area of research on generative flow networks (GFlowNets; Bengio et al., 2021a). GFlowNets have been formulated as a reinforcement learning (RL) algorithm – with states, actions, and rewards – that constructs an object by a sequence of actions so as to make the marginal likelihood of producing an object proportional to its reward. While hierarchical VI is typically used for distributions over real-valued objects, GFlowNets have been successful at approximating distributions over discrete structures for which exact sampling is intractable, such as for molecule discovery (Bengio et al., 2021a), for Bayesian posteriors over causal graphs (Deleu et al., 2022), or as an amortized learned sampler for approximate maximum-likelihood training of energy-based models (Zhang et al., 2022b). Although GFlowNets appear to have different foundations (Bengio et al., 2021b) and applications than hierarchical VI algorithms, we show here that the two are closely connected.

As our main theoretical contribution, we show that special cases of variational algorithms and GFlowNets coincide in their expected gradients. In particular, hierarchical VI (Ranganath et al., 2016b) and nested VI (Zimmermann et al., 2021) are related to the trajectory balance and detailed balance objectives for GFlowNets (Malkin et al., 2022; Bengio et al., 2021b). We also point out the differences between VI and GFlowNets: notably, that GFlowNets automatically perform gradient variance reduction by estimating a marginal quantity (the partition function) that acts as a baseline and allow off-policy learning without the need for reweighted importance sampling.

Our theoretical results are accompanied by experiments that examine what similarities and differences emerge when one applies hierarchical VI algorithms to discrete problems where GFlowNets

---

∗Equal contribution. Contact: `nikolay.malkin@mila.quebec`.

have been used before. These experiments serve two purposes. First, they supply a missing hierarchical VI baseline for problems where GFlowNets have been used in past work. The relative performance of this baseline illustrates the aforementioned similarities and differences between VI and GFlowNets. Second, the experiments demonstrate the ability of GFlowNets, not shared by hierarchical VI, to learn from off-policy distributions without introducing high gradient variance. We show that this ability to learn with exploratory off-policy sampling is beneficial in discrete probabilistic modeling tasks, especially in cases where the target distribution has many modes.

## 2 THEORETICAL RESULTS

### 2.1 GFLOWNETS: NOTATION AND BACKGROUND

We consider the setting of Bengio et al. (2021a). We are given a pointed[1] directed acyclic graph (DAG) $\mathcal{G} = (\mathcal{S}, \mathbb{A})$, where $\mathcal{S}$ is a finite set of vertices (*states*), and $\mathbb{A} \subset \mathcal{S} \times \mathcal{S}$ is a set of directed edges (*actions*). If $s \rightarrow s'$ is an action, we say $s$ is a *parent* of $s'$ and $s'$ is a *child* of $s$. There is exactly one state that has no incoming edge, called the *initial state* $s_0 \in \mathcal{S}$. States that have no outgoing edges are called *terminating*. We denote by $\mathcal{X}$ the set of terminating states. A *complete trajectory* is a sequence $\tau = (s_0 \rightarrow \ldots \rightarrow s_n)$ such that each $s_i \rightarrow s_{i+1}$ is an action and $s_n \in \mathcal{X}$. We denote by $\mathcal{T}$ the set of complete trajectories and by $x_\tau$ the last state of a complete trajectory $\tau$.

GFlowNets are a class of models that amortize the cost of sampling from an intractable target distribution over $\mathcal{X}$ by learning a functional approximation of the target distribution using its unnormalized density or reward function, $R : \mathcal{X} \rightarrow \mathbb{R}^+$. While there exist different parametrizations and loss functions for GFlowNets, they all define a *forward transition probability function*, or a *forward policy*, $P_F(- \mid s)$, which is a distribution over the children of every state $s \in \mathcal{S}$. The forward policy is typically parametrized by a neural network that takes a representation of $s$ as input and produces the logits of a distribution over its children. Any forward policy $P_F$ induces a distribution over complete trajectories $\tau \in \mathcal{T}$ (denoted by $P_F$ as well), which in turn defines a marginal distribution over terminating states $x \in \mathcal{X}$ (denoted by $P_F^\top$):

$$P_F(\tau = (s_0 \rightarrow \ldots \rightarrow s_n)) = \prod_{i=0}^{n-1} P_F(s_{i+1} \mid s_i) \qquad \forall \tau \in \mathcal{T}, \qquad (1)$$

$$P_F^\top(x) = \sum_{\tau \in \mathcal{T} : x_\tau = x} P_F(\tau) \qquad \forall x \in \mathcal{X}. \qquad (2)$$

Given a forward policy $P_F$, terminating states $x \in \mathcal{X}$ can be sampled from $P_F^\top$ by sampling trajectories $\tau$ from $P_F(\tau)$ and taking their final states $x_\tau$.

GFlowNets aim to find a forward policy $P_F$ for which $P_F^\top(x) \propto R(x)$. Because the sum in (2) is typically intractable to compute exactly, training objectives for GFlowNets introduce auxiliary objects into the optimization. For example, the trajectory balance objective (TB; Malkin et al., 2022) introduces an auxiliary *backward policy* $P_B$, which is a learned distribution $P_B(- \mid s)$ over the *parents* of every state $s \in \mathcal{S}$, and an estimated partition function $Z$, typically parametrized as $\exp(\log Z)$ where $\log Z$ is the learned parameter. The TB objective for a complete trajectory $\tau$ is defined as

$$\mathcal{L}_{\text{TB}}(\tau; P_F, P_B, Z) = \left( \log \frac{Z \cdot P_F(\tau)}{R(x_\tau) P_B(\tau \mid x_\tau)} \right)^2, \qquad (3)$$

where $P_B(\tau \mid x_\tau) = \prod_{(s \rightarrow s') \in \tau} P_B(s \mid s')$. If $\mathcal{L}_{\text{TB}}$ is made equal to 0 for every complete trajectory $\tau$, then $P_F^\top(x) \propto R(x)$ for all $x \in \mathcal{X}$ and $Z$ is the inverse constant of proportionality: $Z = \sum_{x \in \mathcal{X}} R(x)$.

The objective (3) is minimized by sampling trajectories $\tau$ from some distribution and making gradient steps on (3) with respect to the parameters of $P_F$, $P_B$, and $\log Z$. The distribution from which $\tau$ is sampled amounts to a choice of scalarization weights for the multi-objective problem of minimizing (3) over all $\tau \in \mathcal{T}$. If $\tau$ is sampled from $P_F(\tau)$ – note that this is a nonstationary scalarization – we say the algorithm runs *on-policy*. If $\tau$ is sampled from another distribution, the algorithm runs *off-policy*; typical choices are to sample $\tau$ from a tempered version of $P_F$ to encourage exploration (Bengio et al., 2021a; Deleu et al., 2022) or to sample $\tau$ from the backward policy $P_B(\tau|x)$ starting from given terminating states $x$ (Zhang et al., 2022b). By analogy with the RL nomenclature, we call the *behavior policy* the one that samples $\tau$ for the purpose of obtaining a stochastic gradient, e.g, the gradient of the objective $\mathcal{L}_{\text{TB}}$ in (3) for the sampled $\tau$.

---

[1]A pointed DAG is one with a designated initial state.

Other objectives have been studied and successfully used in past works, including detailed balance (DB; proposed by Bengio et al. (2021b) and evaluated by Malkin et al. (2022)) and subtrajectory balance (SubTB; Madan et al., 2022). In the next sections, we will show how the TB objective relates to hierarchical variational objectives. In §C, we generalize this result to the SubTB loss, of which both TB and DB are special cases.

## 2.2 Hierarchical variational models and GFlowNets

Variational methods provide a way of sampling from distributions by means of learning an approximate probability density. Hierarchical variational models (HVMs; Ranganath et al., 2016b; Sobolev & Vetrov, 2019; Vahdat & Kautz, 2020; Zimmermann et al., 2021)) typically assume that the sample space is a set of sequences $(z_1, \ldots, z_n)$ of fixed length, with an assumption of conditional independence between $z_{i-1}$ and $z_{i+1}$ conditioned on $z_i$, i.e., the likelihood has a factorization $q(z_1, \ldots, z_n) = q(z_1)q(z_2|z_1) \ldots q(z_n|z_{n-1})$. The marginal likelihood of $z_n$ in a hierarchical model involves a possibly intractable sum,

$$q(z_n) = \sum_{z_1, \ldots, z_{n-1}} q(z_1)q(z_2|z_1) \ldots q(z_n|z_{n-1}).$$

The goal of VI algorithms is to find the conditional distributions $q$ that minimize some divergence between the marginal $q(z_n)$ and a target distribution. The target is often given as a distribution with intractable normalization constant: a typical setting is a Bayesian posterior (used in VAEs, variational EM, and other applications), for which we desire $q(z_n) \propto p_{\text{likelihood}}(x|z_n)p_{\text{prior}}(z_n)$.

**The GFlowNet corresponding to a HVM:** Sampling sequences $(z_1, \ldots, z_n)$ from a hierarchical model is equivalent to sampling complete trajectories in a certain pointed DAG $\mathcal{G}$. The states of $\mathcal{G}$ at a distance of $i$ from the initial state are in bijection with possible values of the variable $z_i$, and the action distribution is given by $q$. Sampling from the HVM is equivalent to sampling trajectories from the policy $P_F(z_{i+1}|z_i) = q(z_{i+1}|z_i)$ (and $P_F(z_1|s_0) = q(z_1)$), and the marginal distribution $q(z_n)$ is the terminating distribution $P_F^\top$.

**The HVM corresponding to a GFlowNet:** Conversely, suppose $\mathcal{G} = (\mathcal{S}, \mathbb{A})$ is a graded pointed DAG[2] and that a forward policy $P_F$ on $\mathcal{G}$ is given. Sampling trajectories $\tau = (s_0 \rightarrow s_1 \rightarrow \ldots \rightarrow s_L)$ in $\mathcal{G}$ is equivalent to sampling from a HVM in which the random variable $z_i$ is the identity of the $(i+1)$-th state $s_i$ in $\tau$ and the conditional distributions $q(z_{i+1}|z_i)$ are given by the forward policy $P_F(s_{i+1}|s_i)$. Specifying an approximation of the target distribution in a hierarchical model with $n$ layers is thus equivalent to specifying a forward policy $P_F$ in a graded DAG.

The correspondence can be extended to non-graded DAGs. Every pointed DAG $\mathcal{G} = (\mathcal{S}, \mathbb{A})$ can be canonically transformed into a graded pointed DAG by the insertion of dummy states that have one child and one parent. To be precise, every edge $s \rightarrow s' \in \mathbb{A}$ is replaced with a sequence of $\ell' - \ell(s)$ edges, where $\ell(s)$ is the length of the *longest* trajectory from $s_0$ to $s$, $\ell' = \ell(s')$ if $s' \notin \mathcal{X}$, and $\ell' = \max_{s'' \in \mathcal{S}} \ell(s'')$ otherwise. This process is illustrated in §A. We thus restrict our analysis in this section, without loss of generality, to graded DAGs.

**The meaning of the backward policy:** Typically, the target distribution is over the objects $\mathcal{X}$ of the last layer of a graded DAG, rather than over complete sequences or trajectories. Any backward policy $P_B$ on the DAG turns an unnormalized target distribution $R$ over $\mathcal{X}$ into an unnormalized distribution over complete trajectories $\mathcal{T}$:

$$\forall \tau \in \mathcal{T} \quad P_B(\tau) \propto R(x_\tau)P_B(\tau \mid x_\tau), \quad \text{with unknown partition function } \hat{Z} = \sum_{x \in \mathcal{X}} R(x). \quad (4)$$

The marginal distribution of $P_B$ over terminating states is equal to $R(x)/\hat{Z}$ by construction. Therefore, if $P_F$ is a forward policy that equals $P_B$ as a distribution over trajectories, then $P_F^\top(x) = R(x)/\hat{Z} \propto R(x)$.

**VI training objectives:** In its most general form, the hierarchical variational objective ('HVI objective' in the remainder of the paper) minimizes a statistical divergence $D_f$ between the learned and the target distributions over trajectories:

$$\mathcal{L}_{\text{HVI},f}(P_F, P_B) = D_f(P_B \| P_F) = \mathbb{E}_{\tau \sim P_F}\left[ f\left(\frac{P_B(\tau)}{P_F(\tau)}\right) \right]. \quad (5)$$

---

[2]We recall some facts about partially ordered sets. A pointed graded DAG is a pointed DAG in which all complete trajectories have the same length. Pointed graded DAGs $\mathcal{G}$ are also characterized by the following equivalent property: the state space $\mathcal{S}$ can be partitioned into disjoint sets $\mathcal{S} = \bigsqcup_{l=0}^{L} \mathcal{S}_l$, with $\mathcal{S}_0 = \{s_0\}$, called *layers*, such that all edges $s \rightarrow s'$ are between states of adjacent layers ($s \in \mathcal{S}_i, s' \in \mathcal{S}_{i+1}$ for some $i$).

Two common objectives are the forward and reverse Kullback-Leibler (KL) divergences (Mnih & Gregor, 2014), corresponding to $f : t \mapsto t \log t$ for $D_{\mathrm{KL}}(P_B \| P_F)$ and $f : t \mapsto -\log t$ for $D_{\mathrm{KL}}(P_F \| P_B)$, respectively. Other $f$-divergences have been used, as discussed in Zhang et al. (2019b); Wan et al. (2020). Note that, similar to GFlowNets, (5) can be minimized with respect to both the forward and backward policies, or can be minimized using a fixed backward policy.

Divergences between two distributions over trajectories and divergences between their two marginal distributions over terminating states distributions are linked via the data processing inequality, assuming $f$ is convex (see e.g. Zhang et al. (2019b)), making the former a sensible surrogate objective for the latter:

$$D_f(R/\hat{Z} \| P_F^\top) \le D_f(P_B \| P_F) \qquad (6)$$

When both $P_B$ and $P_F$ are learned, the divergences with respect to which they are optimized need not be the same, as long as both objectives are 0 if and only if $P_F = P_B$. For example, wake-sleep algorithms (Hinton et al., 1995) optimize the generative model $P_F$ using $D_{\mathrm{KL}}(P_B \| P_F)$ and the posterior $P_B$ using $D_{\mathrm{KL}}(P_F \| P_B)$. A summary of common combinations is shown in Table 1.

Table 1: A comparison of algorithms for approximating a target distribution in a hierarchical variational model or a GFlowNet. The gradients used to update the parameters of the sampling distribution and of the auxiliary backward policy approximate the gradients of various divergences between distributions over trajectories.

| | Surrogate loss | |
| --- | --- | --- |
| Algorithm | $P_F$ (sampler) | $P_B$ (posterior) |
| REVERSE KL | $D_{\mathrm{KL}}(P_F \| P_B)$ | $D_{\mathrm{KL}}(P_F \| P_B)$ |
| FORWARD KL | $D_{\mathrm{KL}}(P_B \| P_F)$ | $D_{\mathrm{KL}}(P_B \| P_F)$ |
| WAKE-SLEEP (WS) | $D_{\mathrm{KL}}(P_B \| P_F)$ | $D_{\mathrm{KL}}(P_F \| P_B)$ |
| REVERSE WAKE-SLEEP | $D_{\mathrm{KL}}(P_F \| P_B)$ | $D_{\mathrm{KL}}(P_B \| P_F)$ |
| On-policy TB | $D_{\mathrm{KL}}(P_F \| P_B)$ | see §2.3 |

We remark that tractable unbiased gradient estimators for objectives such as (5) may not always exist, as we cannot exactly sample from or compute the density of $P_B(\tau)$ when its normalization constant $\hat{Z}$ is unknown. For example, while the REINFORCE estimator gives unbiased estimates of the gradient with respect to $P_F$ when the objective is REVERSE KL (see §2.3), other objectives, such as FORWARD KL, require importance-weighted estimators. Such estimators approximate sampling from $P_B$ by sampling a batch of trajectories $\{\tau_i\}$ from another distribution $\pi$ (which may equal $P_F$) and weighting a loss computed for each $\tau_i$ by a scalar proportional to $\frac{P_B(\tau_i)}{\pi(\tau_i)}$. Such *reweighted importance sampling* is helpful in various variational algorithms, despite its bias when the number of samples is finite (e.g., Bornschein & Bengio, 2015; Burda et al., 2016), but it may also introduce variance that increases with the discrepancy between $P_B$ and $\pi$.

## 2.3 ANALYSIS OF GRADIENTS

The following proposition summarizes our main theoretical claim, relating the GFN objective of (3) and the variational objective of (5). In §C, we extend this result by showing an equivalence between the subtrajectory balance objective (introduced in Malkin et al. (2022) and empirically evaluated in Madan et al. (2022)) and a natural extension of the nested variational objective (Zimmermann et al., 2021) to subtrajectories. A special case of this equivalence is between the Detailed Balance objective (Bengio et al., 2021b) and the nested VI objective (Zimmermann et al., 2021).

**Proposition 1** *Given a graded DAG $\mathcal{G}$, and denoting by $\theta, \phi$ the parameters of the forward and backward policies $P_F, P_B$ respectively, the gradients of the TB objective* (3) *satisfy:*

$$\nabla_\phi D_{\mathrm{KL}}(P_B \| P_F) = \frac{1}{2}\mathbb{E}_{\tau \sim P_B}[\nabla_\phi \mathcal{L}_{\mathrm{TB}}(\tau)], \qquad (7)$$

$$\nabla_\theta D_{\mathrm{KL}}(P_F \| P_B) = \frac{1}{2}\mathbb{E}_{\tau \sim P_F}[\nabla_\theta \mathcal{L}_{\mathrm{TB}}(\tau)]. \qquad (8)$$

The proof of the extended result appears in §C. An alternative proof is provided in §B.

While (8) is the on-policy TB gradient with respect to the parameters of $P_F$, (7) is *not* the on-policy TB gradient with respect to the parameters of $P_B$, as the expectation is taken over $P_B$, not $P_F$. The on-policy TB gradient can however be expressed through a surrogate loss

$$\mathbb{E}_{\tau \sim P_F}[\nabla_\phi \mathcal{L}_{\mathrm{TB}}(\tau)] = \nabla_\phi \left[ D_{\log^2}(P_B \| P_F) + 2(\log Z - \log \hat{Z}) D_{\mathrm{KL}}(P_F \| P_B) \right], \qquad (9)$$

where $\hat{Z} = \sum_{x \in \mathcal{X}} R(x)$, the unknown true partition function. Here $D_{\log^2}$ is the pseudo-$f$-divergence defined by $f(x) = \log(x)^2$, which is not convex for large $x$. (Proof in §B.)

The loss in (7) is not possible to optimize directly unless using importance weighting (cf. the end of §2.2), but optimization of $P_B$ using (7) and $P_F$ using (8) would yield the gradients of REVERSE WAKE-SLEEP in expectation.

**Score function estimator and variance reduction:** Optimizing the reverse KL loss $D_{\text{KL}}(P_F \| P_B)$ with respect to $\theta$, the parameters of $P_F$, requires a likelihood ratio (also known as REINFORCE) estimator of the gradient (Williams, 1992), using a trajectory $\tau$ (or a batch of trajectories), which takes the form:

$$\Delta(\tau) = \nabla_\theta \log P_F(\tau; \theta) c(\tau), \quad \text{where } c(\tau) = \log \frac{P_F(\tau)}{R(x_\tau) P_B(\tau \mid x_\tau)} \tag{10}$$

(Note that the term $\nabla_\theta c(\tau)$ that is typically present in the REINFORCE estimator is 0 in expectation, since $\mathbb{E}_{\tau \sim P_F}[\nabla_\theta \log P_F(\tau)] = \sum_\tau \frac{P_F(\tau)}{P_F(\tau)} \nabla_\theta P_F(\tau) = 0$.) The estimator of (10) is known to exhibit high variance norm, thus slowing down learning. A common workaround is to subtract a baseline $b$ from $c(\tau)$, which does not bias the estimator. The value of the baseline $b$ (also called control variate) that most reduces the trace of the covariance matrix of the gradient estimator is

$$b^* = \frac{\mathbb{E}_{\tau \sim P_F}[c(\tau) \| \nabla_\theta \log P_F(\tau; \theta) \|^2]}{\mathbb{E}_{\tau \sim P_F}[\| \nabla_\theta \log P_F(\tau; \theta) \|^2]},$$

commonly approximated with $\mathbb{E}_{\tau \sim P_F}[c(\tau)]$ (see, e.g., Weaver & Tao (2001); Wu et al. (2018)). This approximation is itself often approximated with a batch-dependent **local** baseline, from a batch of trajectories $\{\tau_i\}_{i=1}^B$:

$$b^{\text{local}} = \frac{1}{B} \sum_{i=1}^B c(\tau_i) \tag{11}$$

A better approximation of the expectation $\mathbb{E}_{\tau \sim P_F}[c(\tau)]$ can be obtained by maintaining a running average of the values $c(\tau)$, leading to a **global** baseline. After observing each batch of trajectories, the running average is updated with step size $\eta$:

$$b^{\text{global}} \leftarrow (1 - \eta) b^{\text{global}} + \eta b^{\text{local}}. \tag{12}$$

This coincides with the update rule of $\log Z$ in the minimization of $\mathcal{L}_{\text{TB}}(P_F, P_B, Z)$ with a learning rate $\frac{\eta}{2}$ for the parameter $\log Z$ (with respect to which the TB objective is quadratic). Consequently, (8) of Prop. 1 shows that the update rule for the parameters of $P_F$, when optimized using the REVERSE KL objective, with (12) as a control variate for the score function estimator of its gradient, is the same as the update rule obtained by optimizing the TB objective using on-policy trajectories.

While learning a backward policy $P_B$ can speed up convergence (Malkin et al., 2022), the TB objective can also be used with a *fixed* backward policy, in which case the REVERSE KL objective and the TB objective differ only in how they reduce the variance of the estimated gradients, if the trajectories are sampled on-policy. In §4, we experimentally explore the differences between the two learning paradigms that arise when $P_B$ is learned, or when the algorithms run off-policy.

## 3 RELATED WORK

**(Hierarchical) VI:** Variational inference (Zhang et al., 2019a) techniques originate from graphical models (Saul et al., 1996; Jordan et al., 2004), which typically include an inference machine and a generative machine to model the relationship between latent variables and observed data. The line of work on black-box VI (Ranganath et al., 2014) focuses on learning the inference machine given a data generating process, i.e., inferring the posterior over latent variables. Hierarchical modeling exhibits appealing properties under such settings as discussed in Ranganath et al. (2016b); Yin & Zhou (2018); Sobolev & Vetrov (2019). On the other hand, works on variational auto-encoders (VAEs) (Kingma & Welling, 2014; Rezende et al., 2014) focus on generative modeling, where the inference machine – the estimated variational posterior – is a tool to assist optimization of the generative machine or decoder. Hierarchical construction of multiple latent variables has also been shown to be beneficial (Sønderby et al., 2016; Maaløe et al., 2019; Child, 2021).

While earlier works simplify the variational family with mean-field approximations (Bishop, 2006), modern inference methods rely on amortized stochastic optimization (Hoffman et al., 2013). One of the oldest and most commonly used ideas is REINFORCE (Williams, 1992; Paisley et al., 2012) which gives unbiased gradient estimation. Follow-up work (Titsias & Lázaro-Gredilla, 2014; Gregor et al., 2014; Mnih & Gregor, 2014; Mnih & Rezende, 2016) proposes advanced estimators to reduce the high variance of REINFORCE. The log-variance loss proposed by Richter et al. (2020) is equivalent in expected gradient of $P_F$ to the on-policy TB loss for a GFlowNet with a batch-optimal value of $\log Z$. On the other hand, path-wise gradient estimators (Kingma & Welling, 2014) have much lower variance, but have limited applicability. Later works combine these two approaches for particular distribution families (Tucker et al., 2017; Grathwohl et al., 2018).

Beyond the evidence lower bound (ELBO) objective used in most variational inference methods, more complex objectives have been studied. Tighter evidence bounds have proved beneficial to the learning of generative machines (Burda et al., 2016; Domke & Sheldon, 2018; Rainforth et al., 2018; Masrani et al., 2019). As KL divergence optimization suffers from issues such as mean-seeking behavior and posterior variance underestimation (Minka, 2005), other divergences are adopted as in expectation propagation (Minka, 2001; Li et al., 2015), more general $f$-divergences (Dieng et al., 2017; Wang et al., 2018; Wan et al., 2020), their special case $\alpha$-divergences (Hernández-Lobato et al., 2016), and Stein discrepancy (Liu & Wang, 2016; Ranganath et al., 2016a). GFlowNets could be seen as providing a novel pseudo-divergence criterion, namely TB, as discussed in this work.

**Wake-sleep algorithms:** Another branch of work, starting with Hinton et al. (1995), proposes to avoid issues from stochastic optimization (such as REINFORCE) by alternatively optimizing the generative and inference (posterior) models. Modern versions extending this framework include reweighted wake-sleep Bornschein & Bengio (2015); Le et al. (2019) and memoised wake-sleep (Hewitt et al., 2020; Le et al., 2022). It was shown in Le et al. (2019) that wake-sleep algorithms behave well for tasks involving stochastic branching.

**GFlowNets:** GFlowNets have been used successfully in settings where RL and MCMC methods have been used in other work, including molecule discovery (Bengio et al., 2021a; Malkin et al., 2022; Madan et al., 2022), biological sequence design (Malkin et al., 2022; Jain et al., 2022; Madan et al., 2022), and Bayesian structure learning (Deleu et al., 2022). A connection of the theoretical foundations of GFlowNets (Bengio et al., 2021a;b) with variational methods was first mentioned by Malkin et al. (2022) and expanded in Zhang et al. (2022a; 2023).

A concurrent and closely related paper (Zimmermann et al., 2022) theoretically and experimentally explores interpolations between forward and reverse KL objectives.

## 4 EXPERIMENTS

The goal of the experiments is to empirically investigate two main observations consistent with the above theoretical analysis:

**Observation 1.** On-policy VI and TB (GFlowNet) objectives can behave similarly in some cases, when both can be stably optimized, while in others on-policy TB strikes a better compromise than either the (mode-seeking) REVERSE KL or (mean-seeking) FORWARD KL VI objectives. This claim is supported by the experiments on all three domains below.

However, in all cases, notable differences emerge. In particular, HVI training becomes more stable near convergence and is sensitive to learning rates, which is consistent with the hypotheses about gradient variance in §2.3.

**Observation 2.** When exploration matters, off-policy TB outperforms both on-policy TB and VI objectives, avoiding the possible high variance induced by importance sampling in off-policy VI. GFlowNets are capable of stable off-policy training without importance sampling. This claim is supported by experiments on all domains, but is especially well illustrated on the realistic domains in §4.2 and §4.3. This capability provides advantages for capturing a more diverse set of modes.

**Observation 1** and **Observation 2** provide evidence that off-policy TB is the best method among those tested in terms of both accurately fitting the target distribution and effectively finding modes, where the latter is particularly important for the challenging molecule graph generation and causal graph discovery problems studied below.

### 4.1 HYPERGRID: EXPLORATION OF LEARNING OBJECTIVES

In this section, we comparatively study the ability of the variational objectives and the GFlowNet objectives to learn a multimodal distribution given by its unnormalized density, or reward function, $R$. We use the synthetic hypergrid environment introduced by Bengio et al. (2021a) and further explored by Malkin et al. (2022). The states form a $D$-dimensional hypergrid with side length $H$, and the reward function has $2^D$ flat modes near the corners of the hypergrid. The states form a pointed DAG, where the source state is the origin $s_0 = \mathbf{0}$, and each edge corresponds to the action of incrementing one coordinate in a state by 1 (without exiting the grid). More details about the environment are provided in §D.1. We focus on the case where $P_B$ is learned, which has been shown to accelerate convergence (Malkin et al., 2022).

In Fig. 1, we compare how fast each learning objective discovers the 4 modes of a $128 \times 128$ grid, with an exploration parameter $R_0 = 0.001$ in the reward function. The gap between the learned distribution $P_F^\top$ and the target distribution is measured by the Jensen-Shannon divergence (JSD)

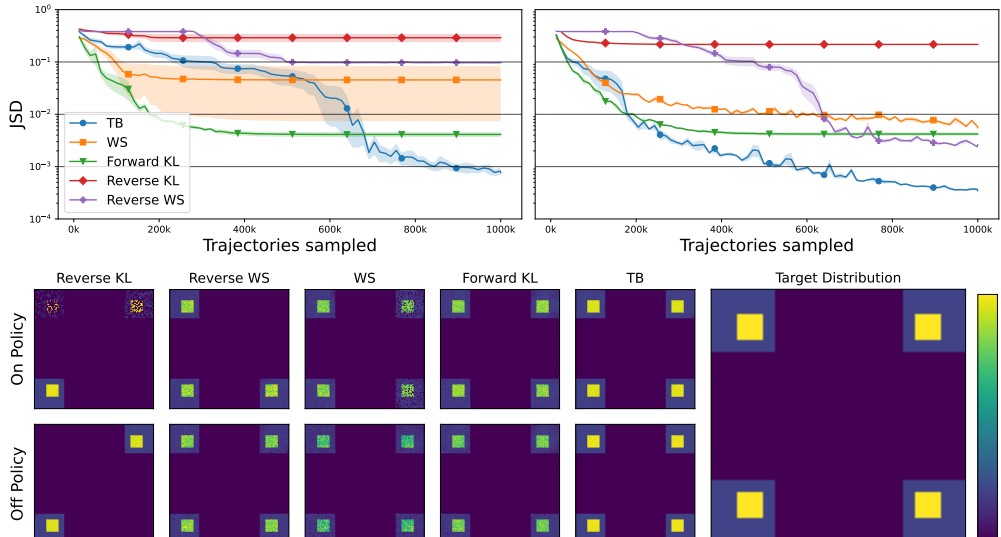

Figure 1: **Top:** The evolution of the JSD between the learned sampler $P_F^\top$ and the target distribution on the $128 \times 128$ grid, as a function of the number of trajectories sampled. Shaded areas represent the standard error evaluated across 5 different runs (on-policy **left**, off-policy **right**). **Bottom:** The average (across 5 runs) final learned distribution $P_F^\top$ for the different algorithms, along with the target distribution. To amplify variation, the plot intensity at each grid position is resampled from the Gaussian approximating the distribution over the 5 runs. Although WS, FORWARD KL, and REVERSE WS (off-policy) find the 4 target modes, they do not model them with high precision, and produce a textured pattern at the modes, where it should be flat.

between the two distributions, to avoid giving a preference to one KL or the other. Additionally, we show graphical representations of the learned 2D terminating states distribution, along with the target distribution. We provide in §E details on how $P_F^\top$ and the JSD are evaluated and how hyperparameters were optimized separately for each learning algorithm.

Exploration poses a challenge in this environment, given the distance that separates the different modes. We thus include in our analysis an off-policy version of each objective, where the behavior policy is different from, but related to, the trained sampler $P_F(\tau)$. The GFlowNet behavior policy used here encourages exploration by reducing the probability of terminating a trajectory at any state of the grid. This biases the learner towards sampling longer trajectories and helps with faster discovery of farther modes. When off-policy, the HVI gradients are corrected using importance sampling weights.

For the algorithms that use a score function estimator of the gradient (FORWARD KL, REVERSE WS, and REVERSE KL), we found that using a global baseline, as explained in §2.2, was better than using the more common local baseline in most cases (see Fig. D.1). This brings the VI methods closer to GFlowNets and thus factors out this issue from the comparison with the GFlowNet objectives.

We see from Fig. 1 that while FORWARD KL and WS – the two algorithms that use $D_{\mathrm{KL}}(P_B \| P_F)$ as the objective for $P_F$ – discover the four modes of the distribution faster, they converge to a local minimum and do not model all the modes with high precision. This is due to the mean-seeking behavior of the forward KL objective, requiring that $P_F^\top$ puts non-zero mass on terminating states $x$ where $R(x) > 0$. Objectives that use the reverse KL to train the forward policy (REVERSE KL and REVERSE WS) are mode-seeking and can thus have a low loss without finding all the modes. The TB GFlowNet objective offers the best of both worlds, as it converges to a lower value of the JSD, discovers the four modes, and models them with high precision. This supports **Observation 1**. Additionally, in support of **Observation 2**, while both the TB objective and the HVI objectives benefit from off-policy sampling, TB benefits more, as convergence is greatly accelerated.

We supplement this study with a comparative analysis of the algorithms on smaller grids in §D.1.

## 4.2 MOLECULE SYNTHESIS

We study the molecule synthesis task from Bengio et al. (2021a), in which molecular graphs are generated by sequential addition of subgraphs from a library of blocks (Jin et al., 2020; Kumar

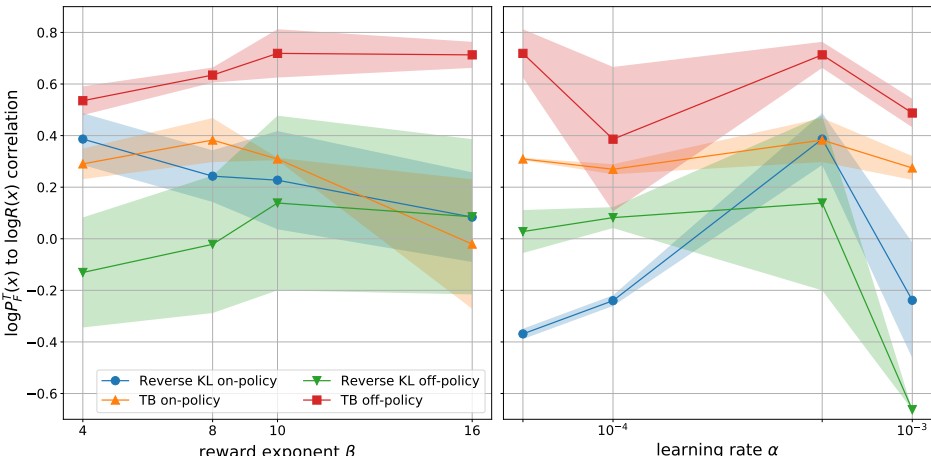

Figure 2: Correlation between marginal sampling log-likelihood and log-reward on the molecule generation task for different learning algorithms, showing the advantage of off-policy TB (red) against on-policy TB (orange) and both on-policy (blue) and off-policy HVI (green). For each hyperparameter setting on the $x$-axis ($\alpha$ or $\beta$), we take the optimal choice of the other hyperparameter ($\beta$ or $\alpha$, respectively) and plot the mean and standard error region over three random seeds.

et al., 2012). The reward function is expressed in terms of a fixed, pretrained graph neural network $f$ that estimates the strength of binding to the soluble epoxide hydrolase protein (Trott & Olson, 2010). To be precise, $R(x) = f(x)^{\beta}$, where $f(x)$ is the output of the binding model on molecule $x$ and $\beta$ is a parameter that can be varied to control the entropy of the sampling model.

Because the number of terminating states is too large to make exact computation of the target distribution possible, we use a performance metric from past work on this task (Bengio et al., 2021a) to evaluate sampling agents. Namely, for each molecule $x$ in a held-out set, we compute $\log P_F^{\top}(x)$, the likelihood of $x$ under the trained model (computable by dynamic programming, see §E), and evaluate the Pearson correlation of $\log P_F^{\top}(x)$ and $\log R(x)$. This value should equal 1 for a perfect sampler, as $\log P_F^{\top}(x)$ and $\log R(x)$ would differ by a constant, the log-partition function $\log \hat{Z}$.

In Malkin et al. (2022), GFlowNet samplers using the DB and TB objectives, with the backward policy $P_B$ fixed to a uniform distribution over the parents of each state, were trained off-policy. Specifically, the trajectories used for DB and TB gradient updates were sampled from a mixture of the (online) forward policy $P_F$ and a uniform distribution at each sampling step, with a special weight depending on the trajectory length used for the termination action.

We wrote an extension of the published code of Malkin et al. (2022) with an implementation of the HVI (REVERSE KL) objective, using a reweighted importance sampling correction. We compare the off-policy TB from past work with the off-policy REVERSE KL, as well as on-policy TB and REVERSE KL objectives. (Note that on-policy TB and REVERSE KL are equivalent in expectation in this setting, since the backward policy is fixed.) Each of the four algorithms was evaluated with four values of the inverse temperature parameter $\beta$ and of the learning rate $\alpha$, for a total of $4 \times 4 \times 4 = 64$ settings. (We also experimented with the off-policy FORWARD KL / WS objective for optimizing $P_F$, but none of the hyperparameter settings resulted in an average correlation greater than 0.1.)

The results are shown in Fig. 2, in which, for each hyperparameter ($\alpha$ or $\beta$), we plot the performance for the optimal value of the other hyperparameter. We make three observations:

- In support of **Observation 2**, off-policy REVERSE KL performs poorly compared to its on-policy counterpart, especially for smoother distributions (smaller values of $\beta$) where more diversity is present in the target distribution. Because the two algorithms agree in the expected gradient, this suggests that importance sampling introduces unacceptable variance into HVI gradients.
- In support of **Observation 1**, the difference between on-policy REVERSE KL and on-policy TB is quite small, consistent with their gradients coinciding in the limit of descent along the full-batch gradient field. However, REVERSE KL algorithms are more sensitive to the learning rate.
- In support of **Observation 2**, off-policy TB gives the best and lowest-variance fit to the target distribution, showing the importance of an exploratory training policy, especially for sparser reward landscapes (higher $\beta$).

Table 2: Comparison of the Jensen-Shannon divergence for Bayesian structure learning, showing the advantage of off-policy TB over on-policy TB and on-policy or off-policy HVI. The JSD is measured between the true posterior distribution $p(G \mid \mathcal{D})$ and the learned approximation $P_F^\top(G)$.

| Objective | Number of nodes | | |
| --- | --- | --- | --- |
| | 3 | 4 | 5 |
| (Modified) Detailed Balance | $5.32 \pm 4.15 \times 10^{-6}$ | $2.05 \pm 0.70 \times 10^{-5}$ | $\mathbf{4.65 \pm 1.08 \times 10^{-4}}$ |
| Off-Policy Trajectory Balance | $\mathbf{3.70 \pm 2.51 \times 10^{-7}}$ | $\mathbf{9.35 \pm 2.99 \times 10^{-6}}$ | $5.44 \pm 2.47 \times 10^{-4}$ |
| On-Policy Trajectory Balance | $0.022 \pm 0.007$ | $0.123 \pm 0.028$ | $0.277 \pm 0.040$ |
| On-Policy REVERSE KL (HVI) | $0.022 \pm 0.007$ | $0.125 \pm 0.027$ | $0.306 \pm 0.042$ |
| Off-Policy REVERSE KL (HVI) | $0.014 \pm 0.008$ | $0.605 \pm 0.019$ | $0.656 \pm 0.009$ |

## 4.3 GENERATION OF DAGS IN BAYESIAN STRUCTURE LEARNING

Finally, we consider the problem of learning the (posterior) distribution over the structure of Bayesian networks, as studied in Deleu et al. (2022). The goal of Bayesian structure learning is to approximate the posterior distribution $p(G \mid \mathcal{D})$ over DAGs $G$, given a dataset of observations $\mathcal{D}$. Following Deleu et al. (2022), we treat the generation of a DAG as a sequential decision problem, where directed edges are added one at a time, starting from the completely disconnected graph. Since our goal is to approximate the posterior distribution $p(G \mid \mathcal{D})$, we use the joint probability $R(G) = p(G, \mathcal{D})$ as the reward function, which is proportional to the former up to a normalizing constant. Details about how this reward is computed, as well as the parametrization of the forward policy $P_F$, are available in §D.3. Note that similarly to §4.2, and following Deleu et al. (2022), we leave the backward policy $P_B$ fixed to uniform.

We only consider settings where the true posterior distribution $p(G \mid \mathcal{D})$ can be computed exactly by enumerating all the possible DAGs $G$ over $d$ nodes (for $d \leq 5$). This allows us to exactly compare the posterior approximations, found either with the GFlowNet objectives or HVI, with the target posterior distribution. The state space grows rapidly with the number of nodes (e.g., there are 29k DAGs over $d = 5$ nodes). For each experiment, we sampled a dataset $\mathcal{D}$ of 100 observations from a randomly generated ground-truth graph $G^\star$; the size of $\mathcal{D}$ was chosen to obtain highly multimodal posteriors. In addition to the (Modified) DB objective introduced by Deleu et al. (2022), we also study the TB (GFlowNet) and the REVERSE KL (HVI) objectives, both on-policy and off-policy.

In Table 2, we compare the posterior approximations found using these different objectives in terms of their Jensen-Shannon divergence (JSD) to the target posterior distribution $P(G \mid \mathcal{D})$. We observe that on the easiest setting (graphs over $d = 3$ nodes), all methods accurately approximate the posterior distribution. But as we increase the complexity of the problem (with larger graphs), we observe that the accuracy of the approximation found with Off-Policy REVERSE KL degrades significantly, while the ones found with the off-policy GFlowNet objectives ((Modified) DB & TB) remain very accurate. We also note that the performance of On-Policy TB and On-Policy REVERSE KL degrades too, but not as significantly; furthermore, both of these methods achieve similar performance across all experimental settings, confirming our **Observation 1**, and the connection highlighted in §2.2. The consistent behavior of the off-policy GFlowNet objectives compared to the on-policy objectives (TB & REVERSE KL) as the problem increases in complexity (i.e., as the number of nodes $d$ increases, requiring better exploration) also supports our **Observation 2**. These observations are further confirmed when comparing the edge marginals $P(X_i \rightarrow X_j \mid \mathcal{D})$ in Fig. D.3 (§D.3), computed either with the target posterior distribution or with the posterior approximations.

## 5 DISCUSSION AND CONCLUSIONS

The theory and experiments in this paper place GFlowNets, which had been introduced and motivated as a reinforcement learning method, in the family of variational methods. They suggest that off-policy GFlowNet objectives may be an advantageous replacement to previous VI objectives, especially when the target distribution is highly multimodal, striking an interesting balance between the mode-seeking (REVERSE KL) and mean-seeking (FORWARD KL) VI variants. This work should prompt more research on how best to choose the behavior policy in off-policy GFlowNet training, seen as a means to efficiently explore and discover modes.

Whereas the experiments performed here focused on the realm of discrete variables, future work should also investigate GFlowNets for continuous action spaces as potential alternatives to VI in continuous-variable domains. We make some first steps in this direction in the Appendix (§F). While this paper was under review, Lahlou et al. (2023) introduced theory for continuous GFlowNets and showed that some of our claims extend to continuous domains.

## AUTHOR CONTRIBUTIONS

N.M., X.J., D.Z., and Y.B. observed the connection between GFlowNets and variational inference, providing motivation for the main ideas in this work. N.M., X.J., and T.D. did initial experimental exploration. S.L., N.M., and D.Z. contributed to the theoretical analysis. S.L. and N.M. extended the theoretical analysis to subtrajectory objectives. D.Z. reviewed the related work. S.L. performed experiments on the hypergrid domain. N.M. performed experiments on the molecule domain and the stochastic control domain. T.D., E.H., and K.E. performed experiments on the causal graph domain. All authors contributed to planning the experiments, analyzing their results, and writing the paper.

## ACKNOWLEDGMENTS

The authors thank Moksh Jain for valuable discussions about the project.

This research was enabled in part by computational resources provided by the Digital Research Alliance of Canada. All authors are funded by their primary institution. We also acknowledge funding from CIFAR, Genentech, Samsung, and IBM.

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

## A  Canonical construction of a graded DAG

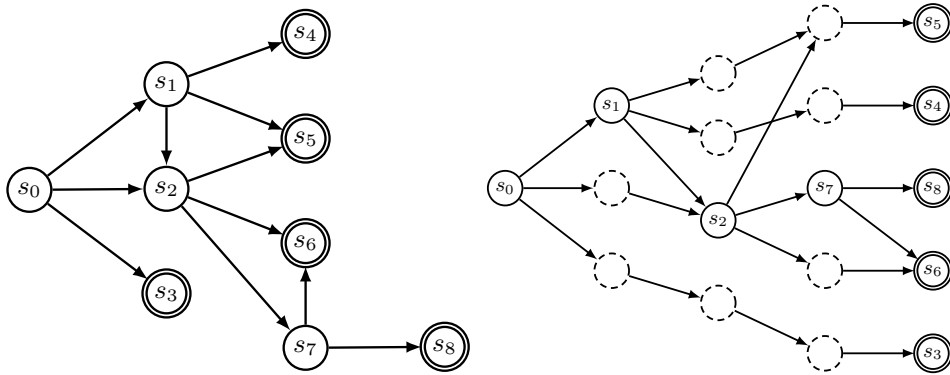

Figure A.1: Illustration of the process by which a DAG (left) can turn into a graded DAG (right). Nodes with a double border represent terminating states. Nodes with a dashed border represent dummy states added to make the DAG graded.

Fig. A.1 shows the canonical conversion of a DAG into a graded DAG as described in §2.2. Note that this operation is idempotent: applying it to a graded DAG yields the same graded DAG.

## B  Proofs

We prove Prop. 1.

**Proof** For a complete trajectory $\tau \in \mathcal{T}$, denote by $c(\tau) = \log \frac{P_F(\tau)}{R(x_\tau)P_B(\tau|x_\tau)}$. We have the following:

$$\nabla_\theta c(\tau) = \nabla_\theta \log P_F(\tau) \tag{13}$$
$$\nabla_\phi c(\tau) = -\nabla_\phi \log P_B(\tau \mid x_\tau) = -\nabla_\phi \log P_B(\tau) \tag{14}$$

Denoting by $f_1 : t \mapsto t \log t$ and $f_2 : t \mapsto -\log t$, which correspond to the forward and reverse KL divergences respectively, and starting from

$$\mathcal{L}_{\mathrm{HVI},f_2}(P_F, P_B) = D_{KL}(P_F \| P_B) = \mathbb{E}_{\tau \sim P_F}\left[\log \frac{P_F(\tau)}{P_B(\tau)}\right] = \mathbb{E}_{\tau \sim P_F}\left[c(\tau)\right] + \log \hat{Z},$$

$$\mathcal{L}_{\mathrm{HVI},f_1}(P_F, P_B) = D_{KL}(P_B \| P_F) = \mathbb{E}_{\tau \sim P_B}\left[\log \frac{P_B(\tau)}{P_F(\tau)}\right] = -\left(\mathbb{E}_{\tau \sim P_B}\left[c(\tau)\right] + \log \hat{Z}\right),$$

we obtain:

$$\nabla_\theta \mathcal{L}_{\mathrm{HVI},f_2}(P_F, P_B) = \nabla_\theta \mathbb{E}_{\tau \sim P_F}\left[c(\tau)\right] = \mathbb{E}_{\tau \sim P_F}\left[\nabla_\theta \log P_F(\tau)c(\tau) + \nabla_\theta c(\tau)\right],$$
$$\nabla_\phi \mathcal{L}_{\mathrm{HVI},f_1}(P_F, P_B) = -\nabla_\phi \mathbb{E}_{\tau \sim P_B}\left[c(\tau)\right] = -\mathbb{E}_{\tau \sim P_B}\left[\nabla_\phi \log P_B(\tau)c(\tau) + \nabla_\phi c(\tau)\right].$$

From (13) and (14), we obtain:

$$\mathbb{E}_{\tau \sim P_F}\left[\nabla_\theta c(\tau)\right] = \mathbb{E}_{\tau \sim P_F}\left[\nabla_\theta \log P_F(\tau)\right] = \sum_{\tau \in \mathcal{T}} P_F(\tau)\nabla_\theta \log P_F(\tau) = \sum_{\tau \in \mathcal{T}} \nabla_\theta P_F(\tau) = \nabla_\theta 1 = 0$$

Hence, for any scalar $Z > 0$, we can write:

$$\mathbb{E}_{\tau \sim P_F}\left[\nabla_\theta c(\tau)\right] = 0 = \mathbb{E}_{\tau \sim P_F}\left[\nabla_\theta \log P_F(\tau)\log Z\right]$$

and similarly

$$\mathbb{E}_{\phi \sim P_F}\left[\nabla_\phi c(\tau)\right] = 0 = \mathbb{E}_{\tau \sim P_B}\left[\nabla_\phi \log P_B(\tau)\log Z\right].$$

Plugging these two equalities back in the HVI gradients above, we obtain:

$$\nabla_\theta \mathcal{L}_{\mathrm{HVI},f_2}(P_F, P_B) = \mathbb{E}_{\tau \sim P_F}\left[\nabla_\theta \log P_F(\tau) \log \frac{ZP_F(\tau)}{R(x_\tau)P_B(\tau \mid x_\tau)}\right]$$

$$\nabla_\phi \mathcal{L}_{\mathrm{HVI},f_1}(P_F, P_B) = -\mathbb{E}_{\tau \sim P_B}\left[\nabla_\theta \log P_B(\tau) \log \frac{ZP_F(\tau)}{R(x_\tau)P_B(\tau \mid x_\tau)}\right]$$

The last two equalities hold for any scalar $Z$ (that does not depend on the parameters of $P_F, P_B$, and that does not depend on any trajectory). In particular, the equations hold for the parameter $Z$ of the Trajectory Balance objective. It thus follows that:

$$\nabla_\theta \mathcal{L}_{\mathrm{HVI},f_2}(P_F, P_B) = \frac{1}{2}\mathbb{E}_{\tau \sim P_F}\left[\nabla_\theta\left(\log \frac{ZP_F(\tau)}{R(x_\tau)P_B(\tau \mid x_\tau)}\right)^2\right] = \frac{1}{2}\mathbb{E}_{\tau \sim P_B}[\nabla_\theta \mathcal{L}_{\mathrm{TB}}(\tau; P_F, P_B, Z)]$$

$$\nabla_\phi \mathcal{L}_{\mathrm{HVI},f_1}(P_F, P_B) = \frac{1}{2}\mathbb{E}_{\tau \sim P_B}\left[\nabla_\theta\left(\log \frac{ZP_F(\tau)}{R(x_\tau)P_B(\tau \mid x_\tau)}\right)^2\right] = \frac{1}{2}\mathbb{E}_{\tau \sim P_B}[\nabla_\phi \mathcal{L}_{\mathrm{TB}}(\tau; P_F, P_B, Z)]$$

As an immediate corollary, we obtain that the expected on-policy TB gradient does not depend on the estimated partition function $Z$. ∎

Next, we will prove the identity (9), which we restate here:

$$\mathbb{E}_{\tau \sim P_F}[\nabla_\phi \mathcal{L}_{\mathrm{TB}}(\tau)] = \nabla_\phi\left[D_{\log^2}(P_B \| P_F) + 2(\log Z - \log \hat{Z})D_{\mathrm{KL}}(P_F \| P_B)\right]. \qquad (15)$$

**Proof** The RHS of (15) equals

$$\nabla_\phi\left[\mathbb{E}_{\tau \sim P_F}\left[\left(\log \frac{P_B(\tau \mid x_\tau)R(x_\tau)}{\hat{Z}P_F(\tau)}\right)^2 + 2(\log Z - \log \hat{Z})\log \frac{P_F(\tau)\hat{Z}}{P_B(\tau \mid x_\tau)R(x_\tau)}\right]\right]$$

$$=\mathbb{E}_{\tau \sim P_F}\left[\nabla_\phi\left(\left(\log \frac{P_B(\tau \mid x_\tau)R(x_\tau)}{\hat{Z}P_F(\tau)}\right)^2 + 2(\log Z - \log \hat{Z})\log \frac{P_F(\tau)\hat{Z}}{P_B(\tau \mid x_\tau)R(x_\tau)}\right)\right]$$

$$=\mathbb{E}_{\tau \sim P_F}\left[2\nabla_\phi \log P_B(\tau \mid x_\tau)\log \frac{P_B(\tau \mid x_\tau)R(x_\tau)}{\hat{Z}P_F(\tau)} - 2(\log Z - \log \hat{Z})\nabla_\phi \log P_B(\tau \mid x_\tau)\right]$$

$$=2\mathbb{E}_{\tau \sim P_F}\left[\nabla_\phi \log P_B(\tau \mid x_\tau)\log \frac{P_B(\tau \mid x_\tau)R(x_\tau)}{ZP_F(\tau)}\right]$$

$$=\mathbb{E}_{\tau \sim P_F}[\nabla_\phi \mathcal{L}_{\mathrm{TB}}(\tau)] \qquad\qquad ∎$$

## C A VARIATIONAL OBJECTIVE FOR SUBTRAJECTORIES

In this section, we extend the claim made in Prop. 1 to connect alternative GFlowNet losses to other variational objectives. Prop. 1 is thus a partial case of Prop. 2. This provides an alternative proof to Prop. 1.

**The detailed balance objective (DB):** The loss proposed in (Bengio et al., 2021b) parametrizes a GFlowNet using its forward and backward policies $P_F$ and $P_B$ respectively, along with a state flow function $F$, which is a positive function of the states, that matches the target reward function on the terminating states. It decomposes as a sum of transition-dependent losses:

$$\forall s{\to}s' \in \mathbb{A} \quad \mathcal{L}_{\mathrm{DB}}(s{\to}s'; P_F, P_B, F) = \left(\log \frac{F(s)P_F(s' \mid s)}{F(s')P_B(s \mid s')}\right)^2, \text{ where } F(s') = R(s') \text{ if } s' \in \mathcal{X}. \qquad (16)$$

**The subtrajectory balance objective (SubTB):** Both the DB and TB objectives can be seen as special instances of the subtrajectory balance objective (Malkin et al., 2022; Madan et al., 2022). Malkin et al. (2022) suggested instead of defining the state flow function $F$ for every state $s$, a state flow function could be defined on a subset of the state space $\mathcal{S}$, called the *hub states*. The loss can be decomposed into a sum of subtrajectory-dependent losses:

$$\forall \tau = (s_1, \ldots, s_n) \in \mathcal{T}^{\mathrm{partial}} \quad \mathcal{L}_{\mathrm{SubTB}}(\tau; P_F, P_B, F) = \left(\log \frac{F(s_1)P_F(\tau)}{F(s_n)P_B(\tau \mid s_t)}\right)^2, \qquad (17)$$

where $P_F(\tau)$ is defined for partial trajectories similarly to complete trajectories (2), $P_B(\tau \mid s) = \prod_{(s\to s')\in\tau} P_B(s \mid s')$, and we again fix $F(x) = R(x)$ for terminating states $x \in \mathcal{X}$). The SubTB objective reduces to the DB objective for subtrajectories of length 1 and to the TB objective for complete trajectories, in which case we use $Z$ to denote $F(s_0)$.

**A variational objective for transitions:** From now on, we work with a graded DAG $\mathcal{G} = (\mathcal{S}, \mathbb{A})$, in which the state space $\mathcal{S}$ is decomposed into *layers*: $\mathcal{S} = \bigsqcup_{l=0}^{L} \mathcal{S}_l$, with $\mathcal{S}_0 = \{s_0\}$ and $\mathcal{S}_L = \mathcal{X}$.

HVI provides a class of algorithms to learn forward and backward policies on $\mathcal{G}$. Rather than learning these policies ($P_F$ and $P_B$) using a variational objective requiring distributions over complete trajectories, nested variational inference (NVI; Zimmermann et al., 2021)), which combines nested importance sampling and variational inference, defines an objective dealing with distributions over transitions, or edges. To this end, it makes use of positive functions $F_k$ of the states $s_k \in \mathcal{S}_k$, for $k = 0, \ldots, L-1$, to define two sets of distributions $\check{p}_k$ and $\hat{p}_k$ over edges from $\mathcal{S}_k$ to $\mathcal{S}_{k+1}$:

$$\hat{p}_k(s_k \rightarrow s_{k+1}) \propto F_k(s_k) P_F(s_{k+1} \mid s_k) \quad \check{p}_k(s_k \rightarrow s_{k+1}) \propto \begin{cases} R(s_L) P_B(s_k \mid s_L) & k = L-1 \\ F_{k+1}(s_{k+1}) P_B(s_k \mid s_{k+1}) & \text{otherwise} \end{cases}. \tag{18}$$

Learning the policies $P_F, P_B$ and the functions $F_k$ is done by minimizing losses of the form:

$$\mathcal{L}_{\text{NVI}}(P_F, P_B, F) = \sum_{k=0}^{L-1} D_f(\check{p}_k \| \hat{p}_k) \tag{19}$$

The positive function $F_k$ plays the same role as the state flow function in GFlowNets (in the DB objective in particular). Before drawing the links between DB and NVI, we first propose a natural extension of NVI to subtrajectories.

## C.1 A VARIATIONAL OBJECTIVE FOR SUBTRAJECTORIES

Consider a graded DAG $\mathcal{G} = (\mathcal{S}, \mathbb{A})$ where $\mathcal{S} = \bigsqcup_{l=0}^{L} \mathcal{S}_l$, $\mathcal{S}_0 = \{s_0\}$, $\mathcal{S}_L = \mathcal{X}$. Amongst the $L + 1$ layers $l = 0, \ldots, L$, we consider $K + 1 \leq L + 1$ special layers, that we call *junction layers*, of which the states are called *hub states*. We denote by $m_0, \ldots, m_K$ the indices of these layers, and we constrain $m_0 = 0$ to represent the layer comprised of the source state only, and $m_K = L$ representing the terminating states $\mathcal{X}$. On each non-terminating junction layer $m_k \neq L$, we define a state flow function $F_k : \mathcal{S}_{m_k} \rightarrow \mathbb{R}_+^*$. Given any forward and backward policies $P_F$ and $P_B$ respectively, consistent with the DAG $\mathcal{G}$, the state flow functions define two sets of distributions $\check{p}_k$ and $\hat{p}_k$ over partial trajectories starting from a state $s_{m_k} \in \mathcal{S}_{m_k}$ and ending in a state $s_{m_{k+1}} \in \mathcal{S}_{m_{k+1}}$ (we denote by $\mathcal{T}_k$ the set comprised of these partial trajectories, for $k = 0 \ldots K - 1$):

$$\forall \tau_k = (s_{m_k} \rightarrow \ldots \rightarrow s_{m_{k+1}}) \in \mathcal{T}_k \quad \hat{p}_k(\tau_k) \propto F_k(s_{m_k}) P_F(\tau_k), \tag{20}$$

$$\forall \tau_k = (s_{m_k} \rightarrow \ldots \rightarrow s_{m_{k+1}}) \in \mathcal{T}_k \quad \check{p}_k(\tau_k) \propto F_{k+1}(s_{m_{k+1}}) P_B(\tau_k \mid s_{m_{k+1}}), \tag{21}$$

where $F_K$ is fixed to the target reward function $R$.

**Lemma 1** *If $\hat{p}_k = \check{p}_k$ for all $k = 0 \ldots K - 1$, then the forward policy $P_F$ induces a terminating state distribution $P_F^\top$ that matches the target unnormalized distribution (or reward function) $R$.*

**Proof** Consider a complete trajectory $\tau = (s_{m_0} \rightarrow \ldots \rightarrow s_{m_1} \rightarrow \ldots \rightarrow \ldots s_{m_2} \rightarrow \ldots \rightarrow \ldots \rightarrow s_{m_K})$. And let $\tau_k = (s_{m_k} \rightarrow \ldots \rightarrow s_{m_{k+1}})$, for every $k < K$.

Denote by $\hat{Z}_k$ and $\check{Z}_k$ the partition functions (constant of proportionality in (18)) of $\hat{p}_k$ and $\check{p}_k$ respectively, for every $k < K$. It is straightforward to see that for every $0 < k < K$:

$$\hat{Z}_{k+1} = \check{Z}_k = \sum_{s_{m_{k+1}} \in \mathcal{S}_{m_{k+1}}} F_{k+1}(s_{m_{k+1}}) \tag{22}$$

$$\prod_{k=0}^{K-1} \hat{p}_k(\tau_k) = \frac{\prod_{k=0}^{K-1} F_k(s_{m_k})}{\prod_{k=0}^{K-1} \hat{Z}_k} P_F(\tau), \tag{23}$$

$$\prod_{k=0}^{K-1} \check{p}_k(\tau_k) = \frac{\prod_{k=0}^{K-1} F_{k+1}(s_{m_{k+1}})}{\prod_{k=0}^{K-1} \check{Z}_k} P_B(\tau \mid s_{m_K}). \tag{24}$$

Because $\hat{p}_k = \check{p}_k$ for all $k = 0 \ldots K - 1$, then both right-hand sides of (23) and (24) are equal. Combining this with (22), we obtain:

$$\forall \tau \in \mathcal{T} \quad \underbrace{\frac{F_0(s_0)}{\hat{Z}_0}}_{=1} P_F(\tau) = \frac{R(x_\tau)}{\sum_{x \in \mathcal{X}} R(x)} P_B(\tau \mid x), \tag{25}$$

which implies the TB constraint is satisfied for all $\tau \in \mathcal{T}$. Malkin et al. (2022) shows that this is a sufficient condition for the terminating state distribution induced by $P_F$ to match the target reward function $R$, which completes the proof. ∎

Similar to NVI, we can use Lemma 1 to define objective functions for $P_F, P_B, F_k$, of the form:

$$\mathcal{L}_{\text{SubNVI},f}(P_F, P_B, F_{0:K-1}) = \sum_{k=1}^{K-1} D_f(\check{p}_k \| \hat{p}_k) \tag{26}$$

Note that the SubNVI objective of (26) matches the NVI objective (Zimmermann et al., 2021) when all layers are junction layers (i.e. $K = L$, and $m_k = k$ for all $k \le L$), and matches the HVI objective of (5) when only the first and last layers are junction layers (i.e. $K = 1$, $m_0 = 0$, and $m_1 = L$).

C.2 AN EQUIVALENCE BETWEEN THE SUBNVI AND THE SUBTB OBJECTIVES

**Proposition 2** *Given a graded DAG $\mathcal{G}$ as in §2.1, with junction layers $m_0 = 0, m_1, \ldots, m_K = L$ as in §C.1. For any forward and backward policies, and for any positive function $F_k$ defined for the hubs, consider $\hat{p}_k$ and $\check{p}_k$ defined in (20) and (21). The subtrajectory variational objectives of (26) are equivalent to the subtrajectory balance objective (17) for specific choices of the $f$-divergences. Namely, denoting by $\theta, \phi$ the parameters of $P_F, P_B$ respectively:*

$$\mathbb{E}_{\tau_k \sim \check{p}_k}[\nabla_\phi \mathcal{L}_{\text{SubTB}}(\tau_k; P_F, P_B, F)] = 2\nabla_\phi D_{f_1}(\check{p}_k \| \hat{p}_k) \tag{27}$$

$$\mathbb{E}_{\tau_k \sim \hat{p}_k}[\nabla_\theta \mathcal{L}_{\text{SubTB}}(\tau_k; P_F, P_B, F)] = 2\nabla_\theta D_{f_2}(\check{p}_k \| \hat{p}_k) \tag{28}$$

*where $F = F_{0:K-1}$, and $f_1 : t \mapsto t \log t$ and $f_2 : t \mapsto -\log t$.*

**Proof** For a subtrajectory $\tau_k = (s_{m_k} \to \ldots \to s_{m_{k+1}}) \in \mathcal{T}_k$, let $c(\tau_k) = \log \frac{F_k(s_{m_k})P_F(\tau_k)}{F_{k+1}(s_{m_{k+1}})P_B(\tau_k|s_{m_{k+1}})}$.

First, note that because $\hat{Z}_k$ and $\check{Z}_k$ are not functions of $\phi, \theta$ ((23)):

$$\nabla_\phi c(\tau_k) = -\nabla_\phi \log \frac{F_{k+1}(s_{m_{k+1}})P_B(\tau_k \mid s_{m_{k+1}})}{\check{Z}_k} = -\nabla_\phi \log \check{p}_k(\tau_k) \tag{29}$$

$$\nabla_\theta c(\tau_k) = \nabla_\theta \log \frac{F_k(s_{m_k})P_F(\tau_k)}{\hat{Z}_k} = \nabla_\phi \log \hat{p}_k(\tau_k) \tag{30}$$

We will prove (27). The proof of (28) follows the same reasoning, and is left as an exercise for the reader.

$$D_{f_1}(\check{p}_k \| \hat{p}_k) = D_{KL}(\check{p}_k \| \hat{p}_k)$$

$$\nabla_\phi D_{f_1}(\check{p}_k \| \hat{p}_k) = \nabla_\phi \sum_{\tau_k \in \mathcal{T}_k} \check{p}_k(\tau_k) \log \frac{\check{p}_k(\tau_k)}{\hat{p}_k(\tau_k)}$$

$$= -\nabla_\phi \sum_{\tau_k \in \mathcal{T}_k} \check{p}_k(\tau_k) c(\tau_k) + \underbrace{\nabla_\phi \log \frac{\hat{Z}_k}{\check{Z}_k}}_{=0, \text{ according to (23)}}$$

$$= -\sum_{\tau_k \in \mathcal{T}_k} (\nabla_\phi \check{p}_k(\tau_k) c(\tau_k) + \check{p}_k(\tau_k) \nabla_\phi c(\tau_k))$$

$$= -\sum_{\tau_k \in \mathcal{T}_k} (\check{p}_k(\tau_k) \nabla_\phi \log \check{p}_k(\tau_k) c(\tau_k) + \check{p}_k(\tau_k) \nabla_\phi c(\tau_k))$$

$$= -\mathbb{E}_{\tau_k \sim \check{p}_k}[\nabla_\phi \log \check{p}_k(\tau_k) c(\tau_k)] + \sum_{\tau_k \in \mathcal{T}_k} \check{p}_k(\tau_k) \nabla_\phi \log \check{p}_k(\tau_k) \quad \text{following (29)}$$

$$= -\mathbb{E}_{\tau_k \sim \check{p}_k}[\nabla_\phi \log P_B(\tau_k \mid s_{m_{k+1}}) c(\tau_k)] + \nabla_\phi \underbrace{\sum_{\tau_k \in \mathcal{T}_k} \check{p}_k(\tau_k)}_{=0}$$

$$= \mathbb{E}_{\tau_k \sim \check{p}_k}\left[\nabla_\phi \log P_B(\tau_k \mid s_{m_{k+1}}) \log \frac{F_{k+1}(s_{m_{k+1}}) P_B(\tau_k \mid s_{m_{k+1}})}{F_k(s_{m_k}) P_F(\tau_k)}\right]$$

$$= \frac{1}{2}\mathbb{E}_{\tau_k \sim \check{p}_k}\left[\nabla_\phi \left(\log \frac{F_k(s_{m_k}) P_F(\tau_k)}{F_{k+1}(s_{m_{k+1}}) P_B(\tau_k \mid s_{m_{k+1}})}\right)^2\right]$$

$$= \frac{1}{2}\mathbb{E}_{\tau_k \sim \check{p}_k}[\nabla_\phi \mathcal{L}_{\text{SubTB}}(\tau_k; P_F, P_B, F)] \qquad \blacksquare$$

As a special case of Prop. 2, when the state flow function is defined for $s_0$ only (and for the terminating states, at which it equals the target reward function), i.e. when $K = 1$, the distribution $\hat{p}_0(\tau)$ and $P_F(\tau)$ are equal, and so are the distributions $\check{p}_0(\tau)$ and $P_B(\tau)$. We thus obtain the first two equations of Prop. 1 as a consequence of Prop. 2.

## D  ADDITIONAL EXPERIMENTAL DETAILS

### D.1  HYPERGRID EXPERIMENTS

**Details about the environment**  For completeness, we provide more details about the environment, as explained in Malkin et al. (2022). In a $D$-dimension hypergrid of side length $H$, the state space $\mathcal{S}$ is partitioned into the non-terminating states $\mathcal{S}^o = \{0, \dots, H-1\}^D$ and terminating states $\mathcal{X} = \mathcal{S}^\top = \{0, \dots, H-1\}^D$. The initial state is $\mathbf{0}_{\mathbb{R}^D} = (0, \dots, 0) \in \mathcal{S}^o$, and in addition to the transitions from a non-terminating state to another (by incrementing one coordinate of the state), an "exit" action is available for all $s \in \mathcal{S}^o$, that leads to a terminating state $s^\top \in \mathcal{S}^\top$. The reward at a terminating state $s^\top = (s^1, \dots, s^D)^\top$ is:

$$R(s^\top) = R_0 + 0.5 \prod_{d=1}^{D} \mathbf{1}\left[\left|\frac{s^d}{H-1} - 0.5\right| \in (0.25, 0.5]\right] + 2 \prod_{d=1}^{D} \mathbf{1}\left[\left|\frac{s^d}{H-1} - 0.5\right| \in (0.3, 0.4]\right], \quad (31)$$

where $R_0$ is an exploration parameter (lower values indicate harder exploration).

**Architectural details**  The forward and backward policies are parametrized as neural networks with 2 hidden layers of 256 units each. The neural networks take as input a one-hot representation of a a state (also called K-hot, or multi-hot representations), which is a $H \times D$ vector including exactly $D$ ones and $(H-1)D$ zeros, and output the logits of $P_F$ and $P_B$ respectively. Forbidden actions (e.g. when a coordinate is already maxed out at $H-1$) are masked out by setting the corresponding logits to $-\infty$ after the forward pass. Unlike Malkin et al. (2022), we do not tie the parameters of $P_F$ and $P_B$.

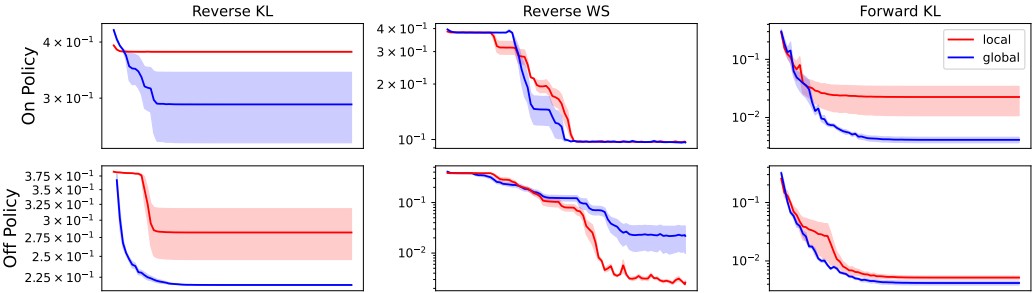

Figure D.1: A comparison of the the type of baseline used (local or global) for the three HVI algorithms that use a score function estimator of the gradient.

**Behavior policy**   The behavior policy is obtained from the forward policy $P_F$ by subtracting a scalar $\epsilon$ from the logits output by the forward policy neural network. The value of $\epsilon$ is decayed from $\epsilon_{init}$ to 0 following a cosine annealing schedule (Loshchilov & Hutter, 2017), and the value $\epsilon = 0$ is reached at an iteration $T_{max}$. The values of $\epsilon_{init}$ and $T_{max}$ were treated as hyperparamters.

**Hyperparameter optimization**   Our experiments have shown that HVI objectives were brittle to the choice of hyperparameters (mainly learning rates), and that the ones used for Trajectory Balance in Malkin et al. (2022) do not perform as well in the larger $128 \times 128$ grid we considered. To obtain a fair comparison between GFlowNets and HVI methods, a particular care was given to the optimization of hyperparameters in this domain. The optimization was performed in two stages:

1. We use a batch size of 64 for all learning objectives, whether on-policy or off-policy, and the Adam optimizer with secondary parameters set to their default values, for the parameters of $P_F$, the parameters of $P_B$, and $\log Z$ (which is initialized at 0). The learning rates of $P_F, P_B, \log Z$, along with a schedule factor $\gamma < 1$ by which they are multiplied when the JSD plateaus for more than 500 iterations (i.e. $500 \times 64$ trajectories sampled), were sought after separately for each combination of learning objective and sampling method (on-policy or off-policy), using a Bayesian search with the JSD evaluated at $200K$ trajectories as an optimization target. The choice of the baseline for HVI methods (except WS, that does not have a score function estimator of the gradient) was treated as a hyperparameter as well.

2. All objectives were then trained for $10^6$ trajectories using all the combinations of hyperparameters found in the first stage, for 5 seeds each. The final set of hyperparameters for each objective and sampling mode was then chosen as the one that leads to the lowest area under the JSD curve (approximated with the trapezoids method).

For off-policy runs, $T_{max}$ was defined as a fraction $1/n$ of the total number of iterations (which is equal to $10^6/64$). The value of $n$ and $\epsilon_{init}$ was optimized the same way as the learning rate and the schedule, as described above.

In Fig. D.1, we illustrate the differences between the two types of baselines considered (**global** and **local**) for the 3 algorithms that use a score function estimator of the gradient, both on-policy and off-policy.

**Smaller environments:**   The environment studied in the main body of text ($128 \times 128$, with $R_0 = 10^{-3}$) already illustrates some key differences between the Forward and Reverse KL objectives. As a sanity check for the HVI methods that failed to converge in this challenging environment, we consider two alternative grids: $64 \times 64$ and $8 \times 8 \times 8 \times 8$, both with an easier exploration parameter ($R_0 = 0.1$), and compare the 5 algorithms on-policy on these two extra domains. Additionally, for the two-dimensional domain ($64 \times 64$), we illustrate in Fig. D.2 a visual representation of the average distribution obtained after sampling $10^6$ trajectories, for each method separately. Interestingly, unlike the hard exploration domain, the two algorithms with the mode-seeking KL (REVERSE KL and REVERSE WS) converge to a lower JSD than the mean-seeking KL algorithms (FORWARD KL and WS), and are on par with TB.

## D.2   MOLECULE EXPERIMENTS

Most experiment settings were identical to those of Malkin et al. (2022), in particular, the reward model $f$ the held-out set of molecules used to compute the performance metric, the GFlowNet model

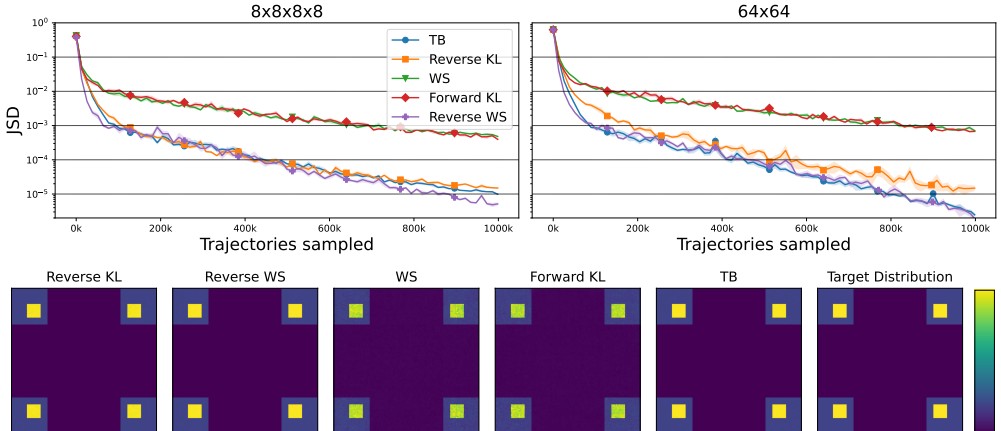

Figure D.2: **Top:** The evolution of the JSD between the learned sampler $P_F^\top$ and the target distribution on the $8 \times 8 \times 8 \times 8$ grid **left** and the $64 \times 64$ grid **right**. Trajectories are sampled on-policy. Shaded areas represent the standard error evaluated across 5 different runs **Bottom:** The average (across 5 runs) final learned distribution $P_F^\top$ for the different algorithms, along with the target distribution. To amplify variation, the plot intensity at each grid position is resampled from the Gaussian approximating the distribution over the 5 runs.

architecture (a graph neural network introduced by by Bengio et al. (2021a)), and the off-policy exploration rate. All models were trained with the Adam optimizer and batch size 4 for a maximum of 50000 batches. The metric was computed after every 5000 batches and the last computed value of the metric was reported, which was sometimes not the value after 50000 batches when the training runs terminated early because of numerical errors.

## D.3 BAYESIAN STRUCTURE LEARNING EXPERIMENTS

**Bayesian Networks** A Bayesian Network is a probabilistic model where the joint distribution over $d$ random variables $\{X_1, \ldots, X_d\}$ factorizes according to a directed acyclic graph (DAG) $G$:

$$p(X_1, \ldots, X_d) = \prod_{i=1}^{d} p(X_i \mid \text{Pa}_G(X_i)),$$

where $\text{Pa}_G(X_i)$ is the set of parent variables of $X_i$ in the graph $G$. Each conditional distribution in the factorization above is also associated with a set of parameters $\theta \in \Theta$. The structure $G$ of the Bayesian Network is often assumed to be known. However, when the structure is unknown, we can learn it based on a dataset of observation $\mathcal{D}$: this is called *structure learning*.

**Structure of the state space** We use the same structure of graded DAG $\mathcal{G}$ as the one described in (Deleu et al., 2022), where each state of $\mathcal{G}$ is itself a DAG $G$, and where actions correspond to adding one edge to the current graph $G$ to transition to a new graph $G'$. Only the actions maintaining the acyclicity of $G'$ are considered valid; this ensures that all the states are well-defined DAGs, meaning that all the states are terminating here (we define a distribution over DAGs). Similar to the hypergrid environment, the action space also contains an extra action "stop" to terminate the generation process, and return the current graph as a sample of our distribution; this "stop" action is denoted $G \to G^\top$, to follow the notation introduced in §2.1.

**Reward function** Our objective in Bayesian structure learning is to approximate the posterior distribution over DAGs $p(G \mid \mathcal{D})$, given a dataset of observations $\mathcal{D}$. Since our goal is to find a forward policy $P_F$ for which $P_F^\top(G) \propto R(G)$ (see §2.1), we can define the reward function as the joint distribution $R(G) = p(G, \mathcal{D}) = p(\mathcal{D} \mid G)p(G)$, where $p(G)$ is a prior over graphs (assumed to be uniform throughout the paper), and $p(\mathcal{D} \mid G)$ is the marginal likelihood. Since the marginal likelihood involves marginalizing over the parameters of the Bayesian Network

$$p(\mathcal{D} \mid G) = \int_{\Theta} p(\mathcal{D} \mid \theta, G)p(\theta \mid G) \, d\Theta,$$

it is in general intractable. We consider here a special class of models, called *linear-Gaussian models*, where the marginal likelihood can be computed in closed form; for this class of models, the log-marginal likelihood is also called the BGe score (Geiger & Heckerman, 1994; Kuipers et al., 2014) in the structure learning literature.

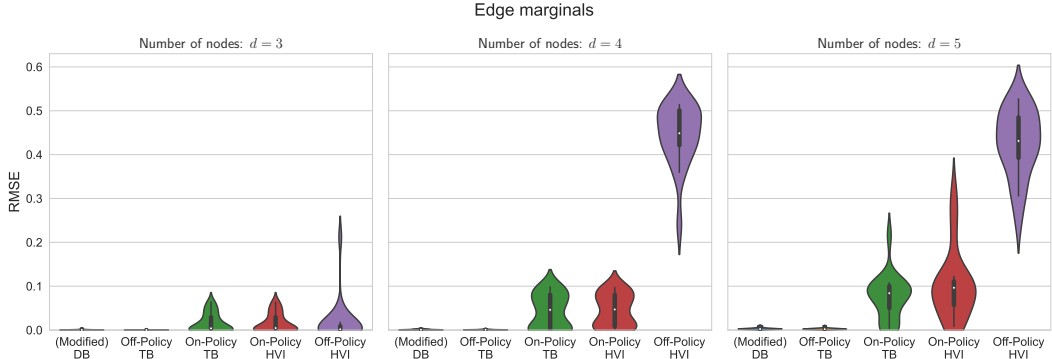

Figure D.3: Comparison of edge marginals computed using the target posterior distribution and using the posterior approximations found either with the GFlowNet objectives, or REVERSE KL. Performance is reported as the Root Mean Square Error (RMSE) between the marginals (lower is better).

For each experiment, we sampled a dataset $\mathcal{D}$ of 100 samples from a randomly generated Bayesian network. The (ground truth) structure of the Bayesian Network was generated following an Erdős-Rényi model, with about $d$ edges on average (to encourage sparsity on such small graphs with $d \le 5$). Once the structure is known, the parameters of the linear-Gaussian model were sampled randomly from a standard Normal distribution $\mathcal{N}(0, 1)$. See (Deleu et al., 2022) for more details about the data generation process. For each setting (different values of $d$) and each objective, we repeated the experiment over 20 different seeds.

**Forward policy**  Deleu et al. (2022) parametrized the forward policy $P_F$ using a linear transformer, taking all the $d^2$ possible edges in the graph $G$ as an input, and returning a probability distribution over those edges, where the invalid actions were masked out. We chose to parametrize $P_F$ using a simpler neural network architecture, based on a graph neural network (Battaglia et al., 2018). The GNN takes the graph $G$ as an input, where each node of the graph is associated with a (learned) embedding, and it returns for each node $X_i$ a pair of embeddings $\boldsymbol{u}_i$ and $\boldsymbol{v}_i$. The probability of adding an edge $X_i \rightarrow X_j$ to transition from $G$ to $G'$ (given that we do not terminate in $G$) is then given by

$$P_F(G' \mid G, \neg G^\top) \propto \exp(\boldsymbol{u}_i^\top \boldsymbol{v}_j),$$

assuming that $X_i \rightarrow X_j$ is a valid action (i.e., it doesn't introduce a cycle in $G$), and where the normalization depends only on all the valid actions. We then use a hierarchical model to obtain the forward policy $P_F(G' \mid G)$, following (Deleu et al., 2022):

$$P_F(G' \mid G) = (1 - P_F(G^\top \mid G))P_F(G' \mid G, \neg G^\top).$$

Recall that the backward policy $P_B$ is fixed here, as the uniform distribution over the parents of $G$ (i.e. all the graphs were exactly one edge has been removed from $G$).

**(Modified) Detailed Balance objective**  For completeness, we recall here the modified Detailed Balance (DB) objective (Deleu et al., 2022) as a special case of the DB objective (Bengio et al., 2021b; see also (16)) when all the states of $\mathcal{G}$ are terminating (which is the case in our Bayesian structure learning experiments):

$$\mathcal{L}_{(M)DB}(G \rightarrow G'; P_F, P_B) = \left(\log \frac{R(G')P_B(G \mid G')P_F(G^\top \mid G)}{R(G)P_F(G' \mid G)P_F(G^\top \mid G)}\right)^2.$$

**Optimization**  Following (Deleu et al., 2022), we used a replay buffer for all our off-policy objectives ((Modified) DB, TB, and REVERSE KL). All the objectives were optimized using a batch size of 256 graphs sampled either on-policy from $P_F$, or from the replay buffer. We used the Adam optimizer, with the best learning rate found among $\{10^{-6}, 3 \times 10^{-6}, 10^{-5}, 3 \times 10^{-5}, 10^{-4}\}$. For the TB objective, we learned $\log Z$ using SGD with a learning rate of 0.1 and momentum 0.8.

**Edge marginals**  In addition to the Jensen-Shannon divergence (JSD) between the true posterior distribution $p(G \mid \mathcal{D})$ and the posterior approximation $P_F^\top(G)$ (see § E for details about how this

divergence is computed), we also compare the edge marginals computed with both distributions. That is, for any edge $X_i \rightarrow X_j$ in the graph, we compare

$$p(X_i \rightarrow X_j \mid \mathcal{D}) = \sum_{G \mid X_i \in \mathrm{Pa}_G(X_j)} p(G \mid \mathcal{D}) \quad \text{and} \quad P_F^\top(X_i \rightarrow X_j) = \sum_{G \mid X_i \in \mathrm{Pa}_G(X_j)} P_F^\top(G).$$

The edge marginal quantifies how likely an edge $X_i \rightarrow X_j$ is to be present in the structure of the Bayesian Network, and is of particular interest in the (Bayesian) structure learning literature. To measure how accurate the posterior approximation $P_F^\top$ is for the different objectives considered here, we use the Root Mean Square Error (RMSE) between $p(X_i \rightarrow X_j \mid \mathcal{D})$ and $P_F^\top(X_i \rightarrow X_j)$, for all possible pairs of nodes $(X_i, X_j)$ in the graph.

Fig. D.3 shows the RMSE of the edge marginals, for different GFlowNet objectives and REVERSE KL (denoted as HVI here for brevity). The results on the edge marginals largely confirm the observations made in §4.3: the off-policy GFlowNet objectives ((Modified) DB & TB) consistently perform well across all experimental settings; On-Policy TB & On-Policy REVERSE KL perform similarly and degrade as the complexity of the experiment increases (as $d$ increases); and Off-Policy REVERSE KL has a performance that degrades the most as the complexity increases, where the edge marginals given by $P_F^\top(X_i \rightarrow X_j)$ do not match the true edge marginals $p(X_i \rightarrow X_j \mid \mathcal{D})$ accurately.

## E  METRICS

**Evaluation of the terminating state distribution $P_F^\top$:**    When the state space is small enough (e.g. graphs with $d \leq 5$ nodes in the Structure learning experiments, or a 2-D hypergrid with length 128, as in the Hypergrid experiments), we can propagate the flows in order to compute the terminating state distribution $P_F^\top$ from the forward policy $P_F$. This is done using a flow function $F$ defined recursively:

$$F(s') = \begin{cases} 1 & \text{if } s' = s_0 \\ \sum_{s \in Par(s')} F(s) P_F(s' \mid s) & \text{otherwise} \end{cases} \tag{32}$$

$P_F^\top$ is then given by:

$$P_F^\top(s^\top) \propto F(s) P_F(s^\top \mid s), \tag{33}$$

The recursion can be carried out by dynamic programming, by enumerating the states in any topological ordering consistent with the graded DAG $\mathcal{G}$. In particular, computation of the flow at a given terminating state $s$ is linear in the number of states and actions that lie on trajectories leading to $s$, and computation of the full distribution $P_F^\top$ is linear in $|\mathcal{S}| + |\mathbb{A}|$.

**Evaluation of the Jensen-Shannon divergence (JSD)**    Similarly, when the state space is small enough, the target distribution $P^\top = R/Z^*$ can be evaluated exactly, given that the marginalization is over $\mathcal{X}$ only. The JSD is a symmetric divergence, thus motivating our choice. The JSD can directly be evaluated as:

$$JSD(P^\top \| P_F^\top) = \frac{1}{2} \left( D_{\mathrm{KL}}(P^\top \| M) + D_{\mathrm{KL}}(P_F^\top \| M) \right) \quad \text{where } M = (P^\top + P_F^\top)/2 \tag{34}$$

$$= \frac{1}{2} \sum_{s \in \mathcal{S}^o} \left( P^\top(s) \log \frac{2 P^\top(s)}{P^\top(s) + P_F^\top(s)} + P_F^\top(s) \log \frac{2 P_F^\top(s)}{P^\top(s) + P_F^\top(s)} \right) \tag{35}$$

## F    EXTENSION TO CONTINUOUS DOMAINS

As a first step towards understanding GFlowNets with continuous action spaces, we perform an experiment on a stochastic control problem. The goal of this experiment is to explore whether the observations in the main text may hold in continuous settings as well.

We consider an environment in which an agent begins at the point $\mathbf{x}_0 = (0, 0)$ in the plane and makes a sequence of $K = 10$ steps over the time interval $[0, 1]$, through points $\mathbf{x}_{0.1}, \mathbf{x}_{0.2}, \ldots, \mathbf{x}_1$. Each step from $\mathbf{x}_t$ to $\mathbf{x}_{t+0.1}$ is Gaussian with learned mean depending on $\mathbf{x}_t$ and $t$ and with fixed variance; the variance is isotropic with standard deviation $\frac{1}{2\sqrt{K}}$. Equivalently, the agent samples the Euler-Maruyama discretization with interval $\Delta t = \frac{1}{K}$ of the Itô stochastic differential equation

$$d\mathbf{x}_t = f(\mathbf{x}_t, t)\, dt + \frac{1}{2} d\mathbf{w}_t, \tag{36}$$

where $\mathbf{w}_t$ is the two-dimensional Wiener process.

The choice of the drift function $f$ determines the marginal density of the final point, $\mathbf{x}_1$. We aim to find $f$ such that this marginal density is proportional to a given reward function, in this case a quantity proportional to the density function of the standard `8gaussians` distribution, shown in Fig. F.2. We scale the distribution so that the modes of the 8 Gaussian components are at a distance of 2 from the origin and their standard deviations are 0.25.

In GFlowNet terms, the set of states is $\mathcal{S} = \{(\mathbf{0}, 0)\} \cup \{(\mathbf{x}, t) : \mathbf{x} \in \mathbb{R}^2, t \in \{0.1, 0.2, \ldots, 1\}\}$. States with $t = 1$ are terminating. There is an action from $(\mathbf{x}, t)$ to $(\mathbf{x}', t')$ if and only if $t' = t + \Delta t$. The forward policy is given by a conditional Gaussian:

$$P_F((\mathbf{x}', t + \Delta t) \mid (\mathbf{x}, t)) = \mathcal{N}\left(\mathbf{x}' - \mathbf{x}; f(\mathbf{x}, t)\Delta t, \left(\frac{\sqrt{\Delta t}}{2}\right)^2\right). \tag{37}$$

We impose a conditional Gaussian assumption on the backward policy as well, i.e.,

$$P_B((\mathbf{x}, t) \mid (\mathbf{x}', t + \Delta t)) = \begin{cases} \mathcal{N}\left(\mathbf{x} - \mathbf{x}'; \mu_B(\mathbf{x}', t + \Delta t)\Delta t, \sigma_B^2(\mathbf{x}', t + \Delta t)\Delta t\right) & t \neq 0 \\ 1 & t = 0 \end{cases}, \tag{38}$$

where $\mu_B$ and $\log \sigma_B^2$ are learned. Notice that all the policies, except the backward policy from time $\frac{1}{K}$ to time 0, now represent probability *densities*; states can have uncountably infinite numbers of children and parents.

We parametrize the three functions $f, \mu_B, \log \sigma_B^2$ as small (two hidden layers, 64 units per layer) MLPs taking as input the position $\mathbf{x}$ and an embedding of the time $t$. Their parameters can be optimized using any of the five algorithms in Table 1 of the main text.[3] Fig. F.1 shows the marginal densities of $\mathbf{x}_t$ (estimated using KDE) for different $t$ in one well-trained model, as well as some sampled points and paths.

In addition to training on policy, we consider exploratory training policies that add Gaussian noise to the mean of each transition distribution. We experiment with adding standard normal noise scaled by $\sigma_{\exp}$, where $\sigma_{\exp} \in \{0, 0.1, 0.2\}$.

Fig. F.2 compares the marginal densities obtained using different algorithms with on-policy and off-policy training. The algorithms that use a forward KL objective to learn $P_B$ – namely, REVERSE WS and FORWARD KL – are not shown because they encounter NaN values in the gradients early in training, even when using a 10× lower learning rate than that used for all other algorithms ($10^{-3}$ for the parameters of $f, \mu_B, \log \sigma_B^2$ and $10^{-1}$ for the $\log Z$ parameter of the GFlowNet).

These results suggest that the observations made for discrete-space GFlowNets in the main text may continue to hold in continuous settings. The first two rows of Fig. F.2 show that off-policy exploration is essential for finding the modes and that TB achieves a better fit to the target distribution. Just as in Fig. 1, although all modes are found by WAKE-SLEEP, they are modeled with lower precision, appearing off-centre and having an oblong shape, which is reflected in the slightly higher MMD.

---

[3]We conjecture (and strongly believe under mild assumptions) but do not prove that the necessary GFlowNet theory continues to hold when probabilities are placed by probability densities; the results obtained here are evidence in support of this conjecture.

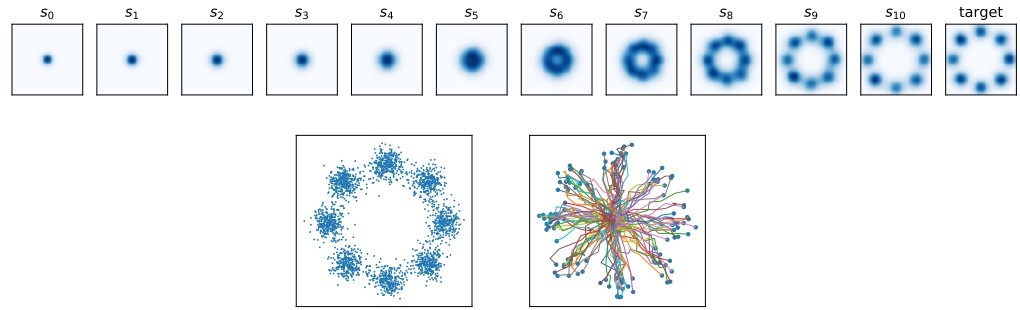

Figure F.1: **Above:** KDE (2560 samples, bandwidth 0.25) of the agent's position after $i$ steps for $i = 0, 1, \ldots, 10$ ($t = 0, 0.1, \ldots, 1$) for a model trained with off-policy TB, showing a close match to the target distribution (also convolved with the KDE kernel for fair comparison). **Below:** A sample of 2560 points from the trained model and the trajectories taken by 128 of the points.

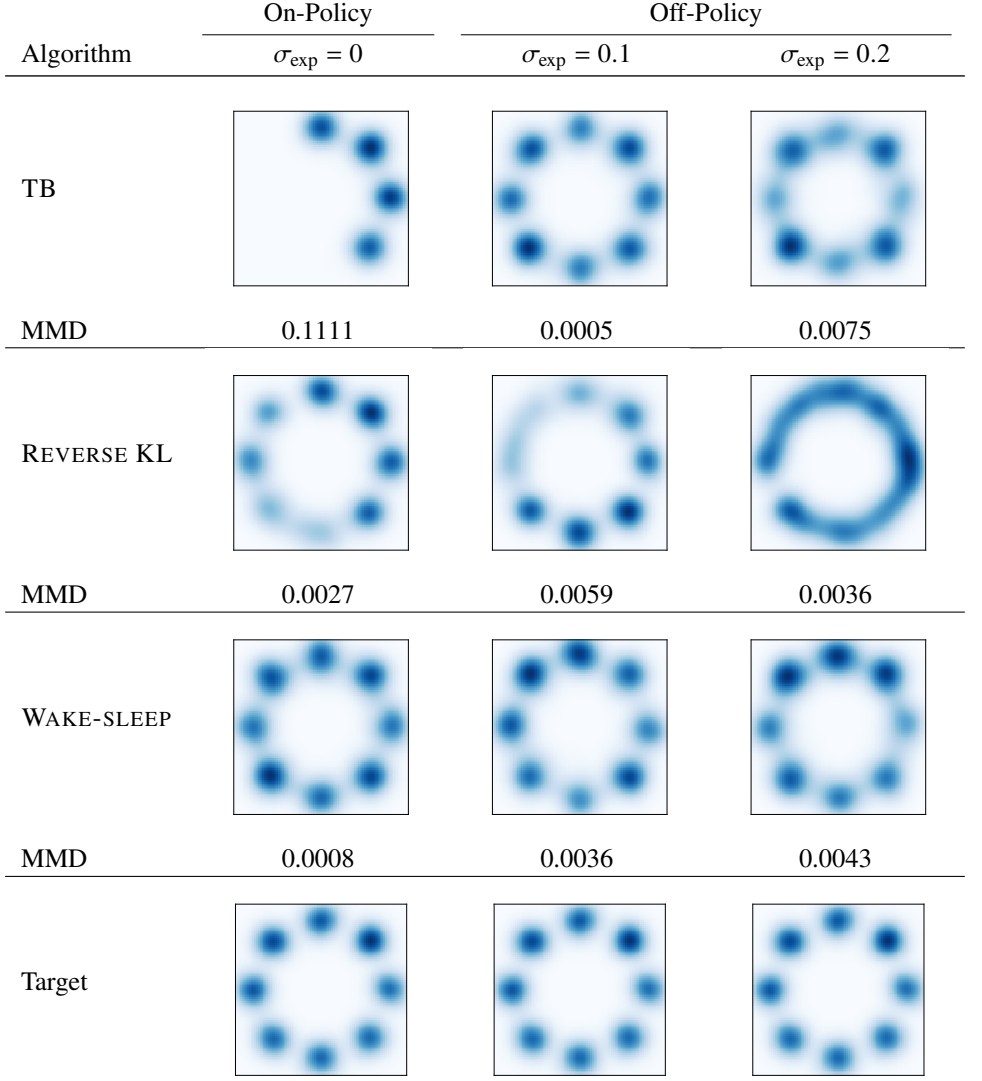

Figure F.2: KDE of learned marginal distributions with various algorithms and exploration policies and MMD with Gaussian kernel $\exp(-\|\mathbf{x} - \mathbf{y}\|^2)$ estimated using 2560 samples.

