# OpenReview forum: "GFlowNets and variational inference"
_ICLR.cc/2023/Conference — ICLR 2023 poster_

### Official Review · Reviewer_taL3 · 2022-10-19

**Confidence:** 4
**Correctness:** 3
**Technical Novelty And Significance:** 4
**Empirical Novelty And Significance:** 4
**Recommendation:** 10

**Clarity, Quality, Novelty And Reproducibility:**

The paper is well-written and can be followed clearly. The discussed connection between the two sets of algorithmic approaches is novel and should lead to interesting new results, especially for continuous domains where the relative performance between them would be an interesting task to explore. The authors are encouraged to further pursue this direction.


### Typos
- p1 last line: hierarhical

**Strength And Weaknesses:**

## Strengths
- The paper demonstrates a novel connection between VI and GFlowNet approaches
- All statements and empirical claims are clearly stated and demonstrated
- A diverse set of experiments in different domains is provided

## Weaknesses
- A, rather minor, weakness is the lack of a stronger evaluation of the connection into the continuous domain, demonstrating the usefulness of the proposed new point of view compared to common VI approaches.


**Summary Of The Paper:**

The authors discuss the relationship between specific families of variational inference approaches (hierarchical variational inference) with generative flow networks as introduced recently by Bengio et al. (2021). The relation refers to equality in the expected gradients of the respective objectives.
The authors evaluate this theoretical relationship further in a set of three experiments of various domains, discussing both similarities and differences.


**Summary Of The Review:**

A well justified, well written, and well evaluated paper.

---

> ### Author Response · Authors · 2022-11-09
> **Authors' response**
>
> Thank you for your comments! We appreciate your support for this paper and are honoured to see such a high score.
>
> > A, rather minor, weakness is the lack of a stronger evaluation of the connection into the continuous domain, demonstrating the usefulness of the proposed new point of view compared to common VI approaches.
>
> We agree that further work on GFlowNets in the continuous domain is a natural next step and, as you mention, would allow additional comparison to common VI approaches. Reviewer 6mPn made a similar note, and we direct you to our response to their review for discussion on this point.
>
> We have also updated the manuscript with an experiment in the continuous domain (Appendix F), which can serve as a starting point for future work.
>
> > p1 last line: hierarhical
>
> Thanks. We have fixed this in the revised version.

---

### Official Review · Reviewer_CLT1 · 2022-10-25

**Confidence:** 3
**Correctness:** 4
**Technical Novelty And Significance:** 3
**Empirical Novelty And Significance:** 3
**Recommendation:** 6

**Clarity, Quality, Novelty And Reproducibility:**

- Clarity/quality: The paper was a bit hard to follow. I think it’d be helpful if the technical exposition could be smoothed out.
- Novelty: The connection between VI algorithms and special cases of GFlowNets is novel and interesting.
- Reproducibility: The submission included code for the DAG and molecule synthesis experiments.


**Strength And Weaknesses:**

Strengths:
- I liked the experiments the authors conducted which compared the empirical behaviors of various training objectives in practice. Understanding the settings in which the objectives differ and figuring out when each method performs better or worse than others is important, and I thought the authors did a good job on this on the synthetic task. It was also nice to see it scaled to slightly harder problems.
- I thought the connection to HVMs and wake-sleep algorithms was interesting, and could open up new research areas along this direction.

Weaknesses:
- The paper was a bit hard to follow at times.


**Summary Of The Paper:**

This paper introduces a connection between GFlowNets and variational inference (VI) algorithms for hierarchical variational models (HVMs), demonstrating that special cases of training HVMs via VI are equivalent to training GFlowNets via the trajectory balance (TB) objective. This is interesting because GFlowNets can be shown to automatically perform variance reduction in gradient estimation for reinforcement learning (RL), showing that GFlowNets may be helpful for RL settings where VI would have been used instead.

**Summary Of The Review:**

The paper provides a theoretical and empirical investigation into the relationship between hierarchical VI and GFlowNets, outlining when they are equivalent and settings when one should be preferred over the other.

---

> ### Author Response · Authors · 2022-11-09
> **Authors' response**
>
> Thank you for your detailed feedback.
>
> > The paper was a bit hard to follow. I think it’d be helpful if the technical exposition could be smoothed out.
>
> We would like the technical exposition to be as clear as possible to all readers and we would be happy to answer questions or revise areas that were confusing. As an example of the changes we could make, we have added a figure in the new Appendix A showing the way in which any DAG is converted to an equivalent graded DAG by the procedure described in the middle of page 3.
>
> We know that Section 2 is the most dense with technical exposition. Our goal was for the reader to understand the necessary background about GFlowNets in Section 2.1, our contributions showing the relationship between the problems solved by GFlowNets and hierarchical variational models in Section 2.2, and our results comparing different learning algorithms in Section 2.3. Were there particular sections or equations that you found unclear?

---

> > ### Author Response · Authors · 2022-12-08
> > **Follow-up**
> >
> > Dear Reviewer CLT1,
> >
> > Thank you again for your review. Since the discussion period is ending soon, we'd like to ask if you have any more suggestions on what parts of the paper you found too dense or what we could do to improve the technical exposition, in addition to the change noted in our response above. We welcome any feedback that will help make the paper more accessible.
> >
> > Thank you,
> >
> > The authors.

---

> > > ### Comment · Reviewer_CLT1 · 2022-12-13
> > > **Thank you!**
> > >
> > > I'd like to thank the authors for their response. After reading the other reviewers' responses, I will keep my score as is.

---

### Official Review · Reviewer_6mPn · 2022-10-26

**Confidence:** 3
**Correctness:** 4
**Technical Novelty And Significance:** 3
**Empirical Novelty And Significance:** 3
**Recommendation:** 6

**Clarity, Quality, Novelty And Reproducibility:**

This paper is well-written and explains clearly the relationship between variational inference and GFlowNet in certain cases. Moreover, this paper conducted plenty of experiments on hypergrid, molecule synthesis, and DAG generation to verify the superiority of the GFlowNet on off-policy training, which should be reproducible with the provided code.

Typo:
A typo in page 3, line 7, q(z_1,...,z_n)=q(z_1)q(z_2|z_1)...q(z_n|z_{n-1}) rather than x_1



**Strength And Weaknesses:**

Strength:
- The connection between the variational inference and GFlowNet is a new finding. This paper finds the intrinsic property, variance reduction, that makes the GFlowNet stand out. They also verify that GFlowNet is capable of stable off-policy training without importance sampling.
- This paper supplies a missing baseline result for hierarchical VI algorithms on modeling discrete random variables.

Weaknesses:
-   As the author mentioned in the conclusion section, this paper only studies the performance of GFlowNet and VI algorithms on modeling discrete random variables (since GFlowNet was mostly studied on such regime), which could be limited considering that VI algorithms are mostly used for continuous variables.




**Summary Of The Paper:**

This paper studies the relationship between variational inference and GFlowNet. This work finds out these two algorithms are equivalent in the sense of the expected gradients of their learning objectives. Moreover, they demonstrate the superiority of the GFlowNet on off-policy training.

**Summary Of The Review:**

Overall this paper is well-written and provides an interesting connection between GFlownet and VI algorithms, and demonstrate the advantage of GFlowNet on off-policy training.

---

> ### Author Response · Authors · 2022-11-09
> **Authors' response**
>
> Thank you for your comments and your thorough reading of the paper.
>
> > this paper only studies the performance of GFlowNet and VI algorithms on modeling discrete random variables (since GFlowNet was mostly studied on such regime), which could be limited considering that VI algorithms are mostly used for continuous variables.
>
> It would indeed be interesting and important to see whether the observations made in this paper transfer to continuous settings. However, a theory for GFlowNets with continuous action spaces is not yet developed, and we think that developing it is a significant enough contribution to warrant its own manuscript.
>
> A sketch of a theory of GFlowNets with continuous action spaces is given in [Bengio et al., 2021b], which proposes to replace policy likelihoods by tractable probability density functions. However, these proposals are given without proof and have never been tested empirically. An alternative approach, which deviates further from the discrete case, was also suggested in a concurrent ICLR submission titled "CFlowNets", which parametrizes policies using *unnormalized* density functions and uses Monte Carlo sampling to estimate policies.
>
> Nevertheless, motivated by your suggestion, we performed a first experiment on a continuous domain, which has been added to the paper. Please see the comment to all reviewers and the new Appendix F.
>
> > Typo: A typo in page 3, line 7, q(z_1,...,z_n)=q(z_1)q(z_2|z_1)...q(z_n|z_{n-1}) rather than x_1
>
> Thanks. We have fixed this in the revised version.

---

> > ### Author Response · Authors · 2022-12-08
> > **Follow-up**
> >
> > Dear Reviewer 6mPn,
> >
> > Thank you again for your review. Since the discussion period is ending soon, we'd like to ask if you have any more questions, such as on the new experiment with continuous variables that we have added to the paper. We look forward to any additional feedback from you.
> >
> > Thank you,
> >
> > The authors.

---

### Author Response · Authors · 2022-11-09
**Continuous experiments and other updates to the paper**

We have just updated the paper, making several changes:
- Fixing some minor errors, including those noted by Reviewers 6mPn and taL3.
- Adding a diagram of the canonical graded DAG construction as Appendix A to improve clarity. (We would be glad to hear any other suggestions on how to improve the technical exposition.)
- Most notably, **an experiment on a domain with continuous action spaces has been added as Appendix F**. This first step towards extending GFlowNets, and our results in particular, to settings that are more standard for variational inference offers promising evidence that the benefits of GFlowNets in off-policy training transfer to continuous domains. Future work can draw connections with related areas, including diffusion modeling and stochastic optimal control.

---

### Author Response · Authors · 2022-11-16
**Questions?**

Dear reviewers,

The response period is ending soon, and we would like to ask if you have any further questions about the continuous-domain experiments that have been added to the paper or anything else we may be able to clarify.

Thank you,

The authors.

---

### Decision · Program_Chairs · 2023-01-20

**Decision:**

Accept: poster

**Justification For Why Not Higher Score:**

This is a nice paper, but two of the reviewers (6mPn, CLT1) do not express enthusiastic support for this paper (only rated the work marginally above the acceptance threshold). The reviewers also did not raise their scores after the rebuttal. The AC believes that Poster is a suitable choice.

**Justification For Why Not Lower Score:**

Based on the overall positive reviews, the AC thinks it deserves a poster presentation.

**Metareview: Summary, Strengths And Weaknesses:**

The paper introduces a connection between variational inference and GFlowNet and evaluates the theoretical relationship in diverse experiments.

Main strength:
+ A novel connection between hierarchical variational inference and generative flow networks
+ Empirical claims are clearly stated and demonstrated in multiple sets of experiments.

Main weaknesses:
- Only studies algorithms on modeling discrete random variables.
- The technical exposition was a bit hard to follow (Reviewer CLT1)

The reviewers appreciate the authors' response. The AC noted the additional experiment on a domain with continuous action spaces in Appendix F. This alleviates the concerns from Reviewer 6mPn.

Overall, all reviewers are positive about this work, the AC recommends to accept this paper.



**Note From Pc:**

if the above contains the word "oral" or "spotlight" please see: "oral" presentation means -> notable-top-5% and "spotlight" means -> notable-top-25%. As stated in our emails, we are disassociating presentation type from AC recommendations

**Summary Of Ac-Reviewer Meeting:**

N/A